DOI: 10.1038/ncomms14707　　OPEN

# MicroRNA-92a is a circadian modulator of neuronal excitability in *Drosophila*

Xiao Chen[1] & Michael Rosbash[1]

Many biological and behavioural processes of animals are governed by an endogenous circadian clock, which is dependent on transcriptional regulation. Here we address post-transcriptional regulation and the role of miRNAs in *Drosophila* circadian rhythms. At least six miRNAs show cycling expression levels within the pigment dispersing factor (PDF) cell-pacemaker neurons; only mir-92a peaks during the night. *In vivo* calcium monitoring, dynamics of PDF projections, ArcLight, GCaMP6 imaging and sleep assays indicate that mir-92a suppresses neuronal excitability. In addition, mir-92a levels within PDF cells respond to light pulses and also affect the phase shift response. Translating ribosome affinity purification (TRAP) and *in vitro* luciferase reporter assay indicate that mir-92a suppresses expression of *sirt2*, which is homologous to human *sir2* and *sirt3*. *sirt2* RNAi also phenocopies mir-92a overexpression. These experiments indicate that *sirt2* is a functional mir-92a target and that mir-92a modulates PDF neuronal excitability via suppressing SIRT2 levels in a rhythmic manner.

---

[1] Department of Biology, Howard Hughes Medical Institute and National Center for Behavioral Genomics, Brandeis University, Waltham, Massachusetts 02454, USA. Correspondence and requests for materials should be addressed to M.R. (email: rosbash@brandeis.edu).

In diverse organisms from bacteria to humans, many biological pathways and processes are controlled by a robust endogenous circadian clock, which is generally entrained by light, food and temperature[1,2]. Rhythmic behaviour in flies is commonly entrained by 12 h light:dark (LD) cycles; these rhythms persist for days and even weeks in constant darkness (DD). Light is a very strong entraining stimulus; therefore, even a brief 10-min light pulse in the middle of the night can robustly shift the phase of the clock.

From the molecular point of view, animal circadian clocks are controlled by a transcription/translation feedback loop. They include in flies four major proteins: TIMELESS (TIM), CLOCK (CLK), CYCLE (CYC) and PERIOD (PER). CLK and CYC heterodimerize to activate transcription of hundreds of genes, including *timeless* (*tim*) and *period* (*per*) by binding to an E-box enhancer element[3,4]. TIM and PER then suppress the activity of the CLK/CYC dimer, leading to a decrease in transcriptional activation[5]. TIM and PER levels decay, and these lower protein levels release CLK/CYC from suppression to begin a new round of transcription. This molecular cycle takes ~24 h and is generally believed to generate the behavioural and physiological cycles that accompany and characterize circadian rhythms.

From the neural circuit point of view, there are ~150 neurons that are particular important for generating rhythmic behaviour in flies. Among them, eight pairs of PDF cells include the key pacemakers[6]. (They are the only neurons in fly brains that express PDF, the neuropeptide pigment-dispersing factor.) This is because PDF cells are critical for circadian rhythms in flies, as they are arrhythmic when these neurons are ablated[7]. PDF cells also dictate the circadian period of the fly in constant darkness, can sense light and regulate the phase shift response[8,9]. These cells are also highly rhythmic from several points of view: (1) their neuronal excitability cycles around the clock[10–12]; (2) PDF dorsal projections undergo daily cycles of remodelling[13]; (3) many more cycling mRNAs were found in PDF cells than in whole heads[14]. This evidence suggests that PDF cells are at the centre of the clock network, orchestrating if not driving the whole system.

Although less studied compared with the transcription/translation feedback loop, post-transcriptional and even post-translational regulation is also important for circadian clock regulation[15–18]. For example, substantial differences were observed between the expression levels of nascent RNA and mRNA in both flies and mice[15,19]. This has kindled our interest in addressing the roles of microRNAs (miRNAs), which are important post-transcriptional regulators[20]. miRNAs generally trigger mRNA degradation and/or translational inhibition by base paring with the 3′ untranslated regions (3′UTR) of mRNAs, leading to a downregulation of gene expression[21]. Only a few miRNAs show cycling expression levels in fly heads[22–27]; therefore, it was of interest to see whether there are more cycling miRNAs in discrete clock neurons like PDF cells, where circadian regulation may be more important for behavioural rhythms and where RNA assays may be less compromised by cell and tissue heterogeneity.

To this end, we quantified expression of 16 miRNAs in PDF cells with reverse transcriptase quantitative PCR (RT–qPCR) and found six cyclers. They include mir-92a, which was the only miRNA that peaked at night. Expression of mir-92a is regulated by both the core clock and light. Our data indicate that mir-92a suppresses neuronal excitability in part by targeting *sirt2* within PDF cells, which strengthens a daily neuronal excitability cycle. Manipulating mir-92a or *sirt2* levels also lead to altered PDF projection morphology and phase shift responses, both of which are known to be linked to neuronal excitability. The effect of mir-92a on neuronal excitability is not restricted to PDF cells, since manipulation of mir-92a levels in sleep- or wake-promoting neurons is sufficient to change sleep duration; at least in dopaminergic neurons this is done via the same mRNA target, *sirt2*.

## Results

**Clock/light regulates mir-92a oscillations in PDF cells**. To identify PDF cell cycling miRNAs, green fluorescent protein (GFP)-labelled PDF cells were manually sorted for RNA extraction and miRNA libraries constructed with an optimized protocol[28]. Although enough miRNA reads were obtained, sample variation was substantial and precluded using the RNA-sequencing data to identify cycling expression patterns; they usually have modest amplitudes (two- to threefold). Therefore, RT–qPCR was used, which produced more consistent results. On the basis of hints (potential cyclers) from the sequencing results, 16 miRNAs were tested, of which 10 showed reproducible daily expression profiles between two biological replicates; 6 of the 10 are rhythmically expressed (Fig. 1a and Supplementary Fig. 1). As only a few miRNAs were found cycling in heads[22–27], the results here suggest more cycling miRNAs in PDF cells than the whole heads.

We focus in this manuscript on mir-92a. It was the only PDF cell cycling miRNA with lower levels during most of the daytime and higher levels during the nighttime (Fig. 1a). In the first day of constant darkness (DD1), miRNA cycling persists but with a 4–8 h phase advance compared to LD, suggesting that light also affects mir-92a expression (Fig. 1b and Supplementary Fig. 2).

To test further whether the cycling expression is under the control of the core molecular clock, the same LD assay was done in *per*[0] flies (*per*[0];*PDF-GAL4;UAS-mCD8::GFP*). They are completely arrhythmic because of a nonsense mutation in *per*, a core clock gene. There is no indication of residual mir-92a cycling in this background, indicating that it is indeed downstream of the core molecular clock (Fig. 1c).

**mir-92a suppresses neuronal excitability**. To address the functions of mir-92a, we first tested whether manipulating mir-92a levels affects the well-characterized circadian morphological changes in the PDF cell termini. They undergo daily fasciculation–defasciculation cycles under circadian control[13]. To this end, mir-92a was overexpressed, or knocked down using a miRNA sponge (SP), in PDF cells with co-expression of mCD8::GFP (*PDF-GAL4;UAS-mCD8::GFP;UAS*-mir-92aOE or *UAS*-mir-92aSP, respectively)[29]. The morphological cycles of the PDF cell termini were quantified with Sholl analysis, that is, assaying the intersections between axon branches and concentric circles (see Methods).

Compared with wild-type (WT) flies (w1118, *PDF-GAL4;UAS-mCD8::GFP/* + in a *w1118* genetic background), overexpression of mir-92a (mir-92aOE, *PDF-GAL4;UAS-mCD8::GFP;UAS*-mir-92aOE) specifically in PDF cells maintains projections in the fasciculated state at both ZT2 and ZT14 as indicated by non-cycling and low numbers of axonal crosses (Fig. 2a). Specifically, overexpression of mir-92a shows an ~37% decrease in maximal axonal crosses at ZT2 and no significant differences in axonal crosses at ZT14 (Fig. 2a and Supplementary Fig. 3). In contrast, the knockdown results in the opposite, namely the defasciculated state, with no significant differences between the control (scramble, *PDF-GAL4;UAS-mCD8::GFP;UAS*-scramble) and knockdown (mir-92aSP, *PDF-GAL4;UAS-mCD8::GFP; UAS*-mir-92aSP) at ZT2 but an ~24% increase in maximal axonal crosses in the mir-92a knockdown at ZT14 (Fig. 2a and Supplementary Fig. 3). No differences of axonal length were observed among genotypes. To address possible developmental effects from use of *PDF-GAL4*, the same experiments were

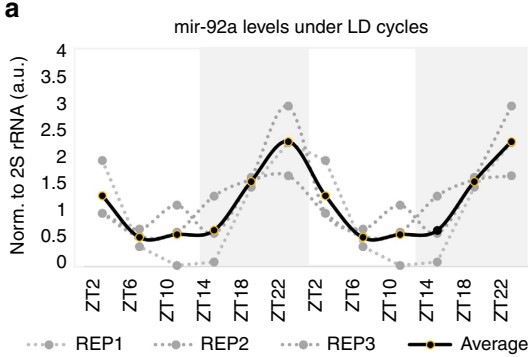

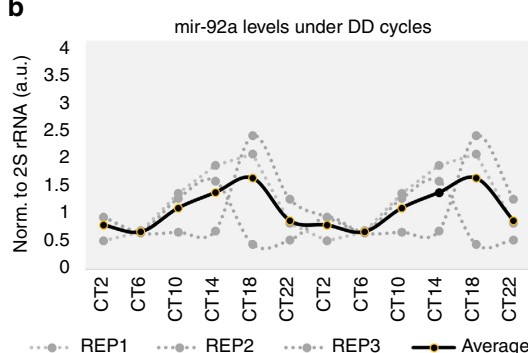

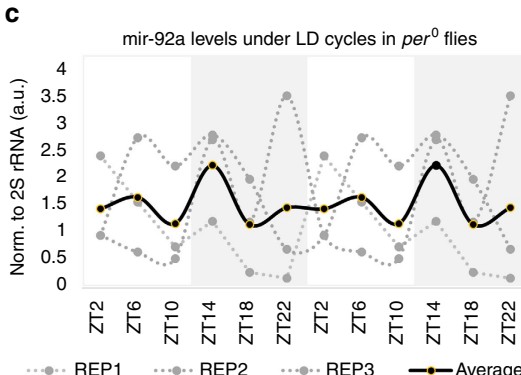

**Figure 1 | mir-92a levels in PDF cells oscillate under both 12 hour LD and DD cycles and are controlled by the molecular clock.** RT–qPCR quantification of mir-92a levels in PDF cells under LD (**a**) or DD (**b**) conditions in WT flies (*PDF-GAL4;UAS-mCD8::GFP*) or *per⁰* (*per⁰; PDF-GAL4;UAS-mCD8::GFP*) mutant flies under LD conditions (**c**). Dotted lines represent the three biological replicates (REP 1, 2 and 3), and the solid line represents the average. Data are double-plotted to show cycling. mir-92a expression levels are normalized to 2S rRNA. a.u. represents arbitrary unit. Grey background indicates the lights-off period and white indicates lights-on.

performed using the inducible geneswitch driver *PDF-GSG*, and the transgenes were only activated in adults[30]. The same results were obtained (Supplementary Fig. 4), indicating that adult-specific manipulation of mir-92a levels changes the fasciculation–defasciculation state of the PDF termini.

Defasciculation can reflect higher neuronal excitability and fasciculation lower excitability[30,31]. Rat mir-92a is also implicated in synaptic scaling[32], suggesting that fly mir-92a suppresses neuronal excitability. To address this possibility more directly, we tested how mir-92a levels affect PDF cell depolarization with high concentration of KCl, by monitoring changes in fluorescence levels of the voltage sensor ArcLight (*PDF-GAL4;UAS-ArcLight;*

UAS-mir-92aOE or UAS-mir-92aSP); they decrease when neurons are depolarized[11]. Brains were attached to the bottom of a chamber with adult haemolymph-like saline (AHL), and baseline fluorescence recorded with a microscope. After 60 s of baseline recording, KCl was perfused into the chamber, which caused an immediate and drastic decrease in fluorescence levels (Fig. 2b and Supplementary Movie 1). Interestingly, mir-92aOE significantly decreased the response, and mir-92aSP consistently but insignificantly increased the response (Fig. 2b).

To further test the effect of mir-92a, nicotine was used to stimulate PDF cells and *PDF-GSG* was used to focus on adult-specific effects. Nicotine is a more physiological agonist and has been shown to fire PDF cells via nicotinic receptors[33]. To this end, the $Ca^{2+}$ sensor GCaMP6 was co-expressed with either UAS-mir-92a or UAS-mir-92aSP under the control of *PDF-GSG*. Although robust increases of fluorescence levels were induced by $3 \times 10^{-6}$ M nicotine, we obtained negative results with the mir-92a manipulations. Changes among genotypes were not statistically significant, perhaps because of relatively large variations among brains as well as the weaker *PDF-GSG* driver compared to *PDF-GAL4* (see below). This was despite a similar trend as the KCl stimulation, namely decreased responses with mir-92a overexpression and increases with mir-92a knockdown (Supplementary Fig. 5).

An independent approach to address mir-92a function was to monitor $Ca^{2+}$ levels in PDF cells using an *in vivo* imaging system. CaLexA is an artificial transcription factor usually located in the cytoplasm when $Ca^{2+}$ levels are low. Higher $Ca^{2+}$ levels cause CaLexA to translocate to the nucleus, where it can bind to the upstream LexAop element and activate luciferase expression of a transgene. Luciferase levels therefore positively correlate with $Ca^{2+}$ levels in this system[34]. To test whether mir-92a levels change $Ca^{2+}$ levels as another proxy of neuronal excitability, luciferase levels were measured every hour for three consecutive days in living WT flies and in flies with manipulated levels of mir-92a (*PDF-GAL4;UAS-CaLexA;LexAop-luciferase;UAS-mir-92aOE* or *UAS-mir-92aSP*).

$Ca^{2+}$ levels in WT flies show a rhythmic pattern with peaks in the morning, consistent with previously reported electrophysiological assays and GCaMP6 imaging of PDF cells (Fig. 2c)[10,12]. Compared to WT, overexpressing mir-92a significantly lowers the luciferase levels, especially during the light period (LP) when mir-92a levels are low (Fig. 2c). On the contrary, knocking down mir-92a increases luciferase levels, especially during the dark period when mir-92a levels are high (Fig. 2c).

GCaMP6 was also used to monitor $Ca^{2+}$ levels in PDF cells. Flies that expressed *UAS-GCaMP6f* together with either *UAS-mir-92aOE* or *UAS-mir-92aSP* under *PDF-GAL4* control were dissected, and the fluorescence levels of their PDF termini measured and quantified. Whereas PDF termini show apparent spontaneous activity, we were not able to accurately estimate the spiking rates and amplitudes from the $Ca^{2+}$ signals because of noise and heterogeneity within individual neurons. We could however quantify the differences between baseline fluorescence levels among genotypes. Consistent with the *in vivo* CaLexA results, knocking down mir-92a levels resulted in an ∼224% increase in baseline fluorescence levels at ZT18–22, and overexpression resulted in an ∼40.6% decrease at ZT6–10 (Supplementary Fig. 6). Since higher $Ca^{2+}$ levels are associated with higher neuronal excitability, these GCaMP6 data as well as the CaLexA data support the hypothesis that mir-92a suppresses neuronal excitability.

**mir-92a levels in sleep-regulating neurons affects sleep**. To test whether the effect of mir-92a on neuronal excitability is restricted to PDF cells, mir-92a levels were either up- or downregulated in

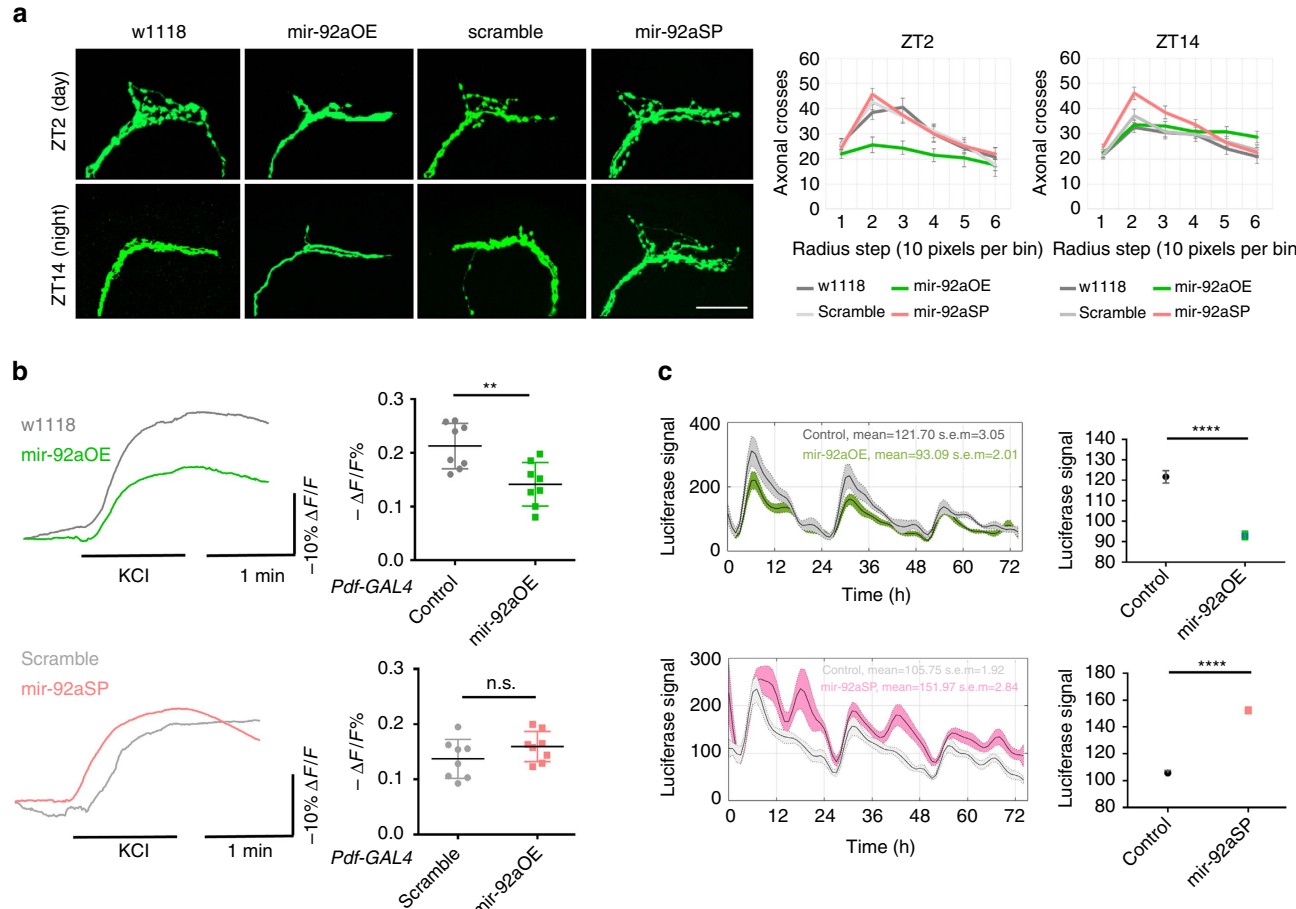

**Figure 2 | Manipulation of mir-92a levels affects neuronal excitability in PDF cells. (a)** Immunostaining of PDF cell projections with anti-GFP antibody at ZT2 or ZT14. mir-92aOE indicates *PDF-GAL4;UAS-mCD8::GFP;UAS*-mir-92aOE flies, and its corresponding control (w1118) is *PDF-GAL4;UAS-mCD8::GFP/ +* (*UAS*-mir-92aOE was backcrossed six times to the w1118 genetic background). mir-92aSP indicates *PDF-GAL4;UAS-mCD8::GFP;UAS*-mir-92aSP flies. It is compared to scramble (*PDF-GAL4;UAS-mCD8::GFP;UAS*-scramble) as a control. SP represents the mir-92a sponge for the knockdown[29]. Flies were entrained for at least 3 days under LD conditions prior to the assay. The left panels show representative images of PDF cell projections of the indicated genotype at the indicated time. Scale bar, 25 μm. Right panels show the quantification performed with Sholl analysis in FIJI. Sixty concentric circles spaced 1 pixel apart were centred on the dorsal ramification point with a radius of 60 pixels (covering the whole or most of the projections). Each radius step is a bin of 10 pixels. Statistical analysis of the quantification is shown in Supplementary Fig. 3. $N = 14$ hemispheres. Error bars represent ± s.e.m. **(b)** ArcLight imaging with high-concentration KCl stimulation in PDF cells. Flies expressing ArcLight in PDF cells (*PDF-GAL4;UAS-ArcLight*) in addition to mir-92a manipulation were imaged for fluorescence-level changes with KCl perfusion at 60 s and wash-out at 115 s indicated by the black bars. The experiments were performed between ZT6–8. Left panels show the average responses of eight PDF neurons (cell bodies of l-LNvs in eight brains) of the indicated genotypes. Maximal changes of $-\Delta F/F\%$ were quantified and used for statistical analysis (right). $N = 8$. Error bars represent ± s.d., **$P < 0.01$, n.s. represents non-significant, two-tailed *t*-test. Figures shown are representative of three trials. **(c)** CaLexA *in vivo* monitoring. Flies expressing the CaLexA transgene were monitored every hour for three consecutive days under LD cycles. Control at the top panel indicates *PDF-GAL4;UAS-CaLexA; LexAop-luciferase/ +* (w1118) and is used to compare with *PDF-GAL4;UAS-CaLexA;LexAop-luciferase/UAS*-mir-92aOE. Control at the bottom panel indicates *PDF-GAL4;UAS-CaLexA;LexAop-luciferase/UAS*-scramble and is compared to *PDF-GAL4;UAS-CaLexA;LexAop-luciferase/UAS*-mir-92aSP. Quantified to the right is the mean of the luciferase signals. $N = 12$. Error bars represent ± s.e.m., ****$P < 0.0001$, two-tailed *t*-test. Figures shown are representative of three trials.

neurons regulating fly sleep. Dopaminergic neurons are well-characterized wake-promoting neurons in mammals and in flies[35]. Overexpression of mir-92a in these neurons (*TH-GAL4;UAS*-mir-92aOE) might lower their excitability and therefore increase sleep. Indeed, sleep duration is significantly increased in the overexpression flies, and sleep duration is significantly reduced when mir-92a is knocked down (Fig. 3a). Similar changes in sleep duration were observed when mir-92a levels were manipulated in another set of wake-promoting neurons driven by *Dvpdf-GAL4* (Supplementary Fig. 7A)[8]. As a negative control, sleep duration of *UAS*-mir-92aOE/ + or *UAS*-mir-92aSP/ + flies without GAL4 drivers was measured and was indistinguishable from WT flies (Supplementary Fig. 7B).

A subset of dorsal clock neurons contain 8–10 cells per brain and are sleep-promoting; the driver is *PDFR-GAL4* from Janelia Research Campus[34]. Overexpression of mir-92a in these cells led to a significant decrease in sleep duration, consistent with the neuronal excitability hypothesis (Fig. 3b). Although mir-92a knockdown in these cells had no effect (Fig. 3b), there are many possible reasons for a negative result, for example, the *GAL4* driver is not sufficiently strong, or endogenous mir-92a is not well expressed in these neurons.

We also assayed sleep duration in mir-92a null (mir-92aKO) flies. It is affected and in the 'correct' direction: the mutant flies show ∼17% less total sleep and ∼44% sleep loss during the LP (Fig. 3c). These sleep results taken together further confirm that

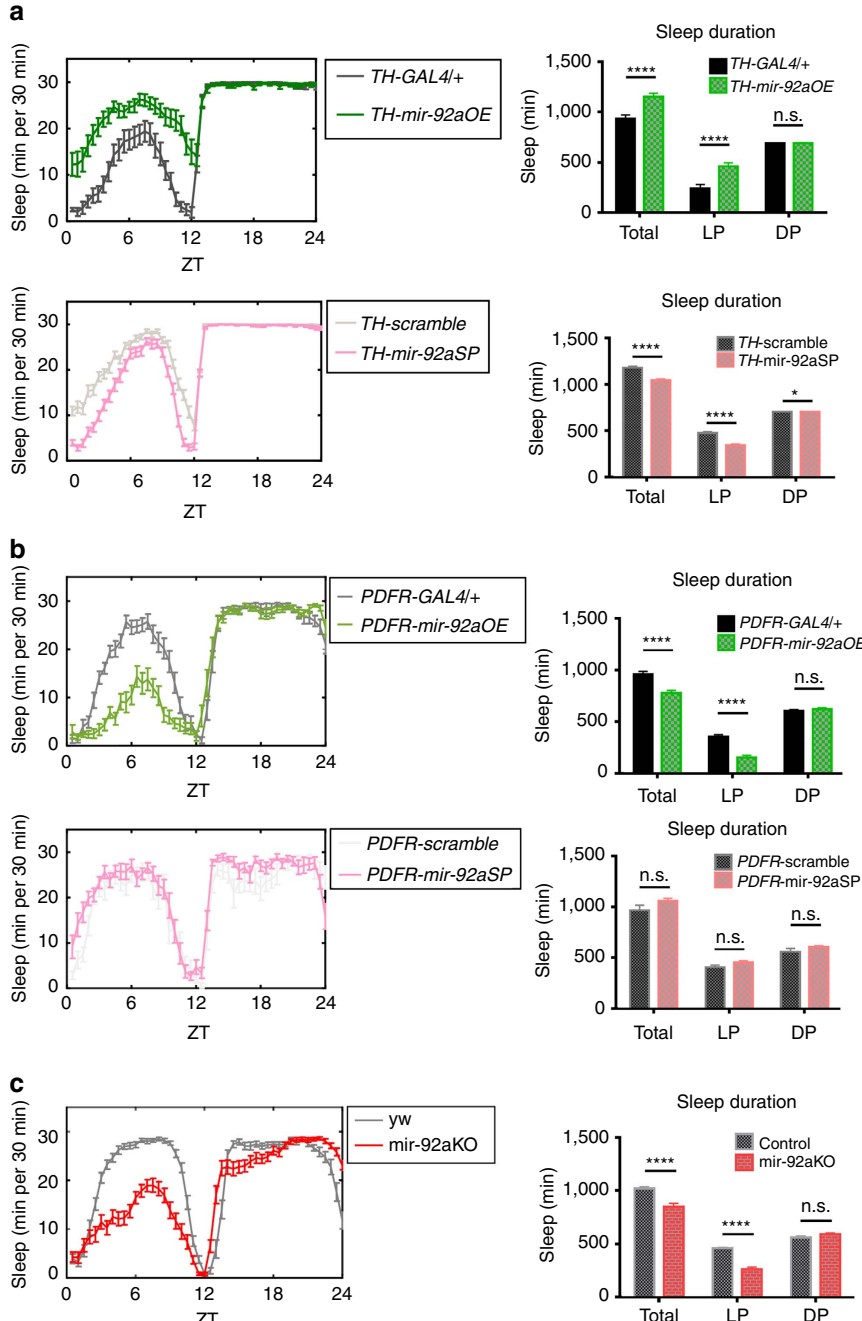

**Figure 3 | Manipulation of mir-92a levels in sleep-regulating neurons affects sleep duration.** Sleep profiles of female flies entrained under LD cycles. mir-92a levels were manipulated in either (**a**) wake-promoting neurons (*TH-GAL4*) or (**b**) sleep-promoting neurons (*PDFR-GAL4*). (**c**) mir-92a null (mir-92aKO) was compared to the WT control with an identical genetic background (yw flies). Sleep duration is quantified to the right. Total indicates total sleep duration (ZT0–24), LP indicates light period (ZT0–12) and DP indicates dark period (ZT12–24). $N = 16$–32. Error bars represent ± s.e.m. n.s. represents non-significant, *$P < 0.05$, ****$P < 0.0001$, two-way ANOVA.

mir-92a suppresses neuronal excitability and also indicate that this regulation is not restricted to PDF cells.

**mir-92a and light-induced phase shifts**. Since mir-92a suppresses neuronal excitability, we wanted to test whether neuronal excitability also affects mir-92a levels. This possibility might also be relevant to the circadian cycling of mir-92a levels. As light pulses during the nighttime fire PDF cells and phase shift the circadian clock, an effect of light/excitability on mir-92a levels could be via an effect on the core clock[8]. We therefore subjected

entrained flies to a 10-min light pulse at either ZT15 or ZT21 and assayed PDF cell mir-92a levels after a subsequent 50 min in the dark. (The same 10 min protocol is used for traditional phase shift assays, with maximum phase delays observed at ZT15 and maximum phase advances at ZT21.)

mir-92a levels increased by approximately twofold in PDF cells after the ZT15 light pulse, and they decreased by ~2.5-fold after the ZT21 light pulse (Fig. 4a). The results appear specific for PDF cells: the light pulses had no effects on mir-92a levels in dopaminergic neurons, which are not activated by light (Supplementary Fig. 8).

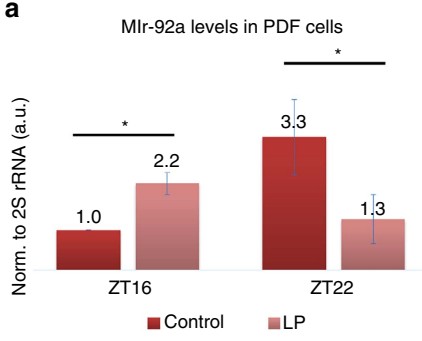

**a**

Mlr-92a levels in PDF cells

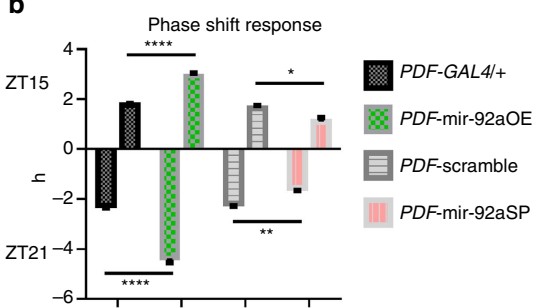

**b**

**Figure 4 | Light pulses at night alter mir-92a levels in PDF cells and manipulation of mir-92a levels affects the phase shift response.**
(**a**) RT–qPCR quantification of mir-92a levels in PDF cells. Flies were entrained for at least 3 days under LD before exposure to a 10-min light pulse at either ZT15 or ZT21. PDF cells were sorted and extracted for RNA at either ZT16 or ZT22, that is, after 50 more minutes in the dark. Control indicates no exposure to light and LP indicates light pulse exposure. $N = 4$. Error bars represent $\pm$ s.e.m. $^*P < 0.05$, two-tailed $t$-test. (**b**) Phase shift response. Ten minutes of dim light ($0.69\,\mathrm{mW\,cm^{-2}}$) exposure was given to flies of the indicated genotypes. Phase differences were quantified between flies subjected to light pulses or not. PDF-GAL4;UAS-mir-92aOE is compared to PDF-GAL4/+ (w1118 background); PDF-GAL4;UAS-mir-92aSP is compared to PDF-GAL4;UAS-scramble. $N = 16$. Error bars represent $\pm$ s.e.m., $^*P < 0.05$, $^{**}P < 0.01$, $^{****}P < 0.0001$, two-way ANOVA. Figures shown are representative of three trails.

To test whether manipulating mir-92a levels in PDF cells has an effect on behaviour, circadian rhythms, sleep and phase shift responses of the mir-92aOE or SP flies (PDF-GAL4; UAS-mir-92aOE or UAS-mir-92aSP) were assayed. There was no effect on rhythmic strength or circadian period (Supplementary Fig. 9A), reflecting perhaps the more modest effect on excitability compared to previous experiments[36,37]. However, sleep and phase shift responses were altered. Knockdown of mir-92a in PDF cells resulted in decreases of sleep duration in both the LP and dark period, whereas overexpression decreases and broadens evening peak activity (Supplementary Fig. 9B). As the role(s) of PDF cells in regulating sleep and circadian rhythms is currently enigmatic, it is unclear how to interpret these changes. However, PDF cells have a more straightforward relationship to phase-shifting[8]. Here dimmer light ($0.69\,\mathrm{mW\,cm^{-2}}$) was used to ensure that the system was not saturated by strong light[38]. Surprisingly perhaps, mir-92aOE leads to bigger shifts both at ZT15 and ZT21, and mir-92aSP causes smaller shifts (Fig. 4b). This observation is consistent with an effect of mir-92a levels on neuronal excitability, with the direction of the effects possibly due to a more labile or sensitive circadian clock when the neuronal excitability of PDF cells is decreased by mir-92aOE and the opposite by the mir-92aSP.

**mir-92a suppresses *sirt2* translation but not mRNA levels.** To identify a mir-92a target responsible for the observed phenotypes, translating ribosome affinity purification (TRAP) was performed on flies with mir-92a either up- or downregulated throughout the circadian system (Tim-GAL4;UAS-RiboTag;UAS-mir-92aOE compared to Tim-GAL4;UAS-RiboTag/ + (control), and Tim-GAL4; UAS-RiboTag;UAS-mir-92aSP compared to Tim-GAL4;UAS-RiboTag/UAS-scramble (control)). Potential targets should show increased mRNA levels (indicated by the Input) and/or translating mRNA levels (indicated by the immunoprecipitation (IP)/Input) with mir-92a downregulation, and decreased mRNA levels with mir-92a upregulation. In addition, these targets should have predicted mir-92a-binding sites in their 3′UTRs with TargetScan (http://www.targetscan.org/). There were 26 genes from our TRAP data that met these criteria (Supplementary Table 1). We then tested these potential candidates with RNA interference (RNAi) in the CaLexA system, the sleep assay and the PDF projection morphology to focus on candidates with the same phenotype as mir-92aOE flies.

*sirt2*, a homologue of mammalian *sir2* and *sirt3* was the only candidate that met all these criteria (see below). It is an NAD-dependent deacetylase of the Sirtuin family. There is one predicted mir-92a-binding site in the *sirt2* 3′UTR. The site is conserved among *Drosophila* species but does not exist in mammals according to TargetScan. The TRAP results indicate no detectable changes in *sirt2* mRNA input levels between WT and mir-92a OE or knockdown flies, whereas the IP/Input levels anticorrelate with mir-92a levels. The data therefore suggest that mir-92a suppresses *sirt2* expression by inhibiting its translation without markedly affecting mRNA stability (Fig. 5a). Unfortunately, no good antibody against fly SIRT2 is available for western blots or immunostaining.

To confirm that mir-92a suppresses *sirt2* expression by binding to the predicted site in the 3′UTR, a luciferase reporter assay was performed in S2 cells. Either a WT or a binding site-mutated *sirt2* 3′UTR was inserted into the psiCHECK2 vector downstream of the *renilla* reporter gene. An internal *firefly luciferase* reporter was expressed separately from the vector as a transfection control (Supplementary Fig. 10).

Co-transfection of *Ub-GAL4* and *UAS-mir-92a* together with the reporter plasmid carrying WT 3′UTR results in a significantly lower renilla/firefly luciferase bioluminescence ratio compared to controls in which the irrelevant gene *dsRed* or the irrelevant miRNA mir-184 was co-transfected instead of mir-92a (Fig. 5b). Moreover, a mutation of the mir-92a-binding site in the 3′UTR eliminates the mir-92a suppression (Fig. 5b). The results confirm that mir-92a suppresses *sirt2* expression both *in vitro* and *in vivo*.

***Sirt2* RNAi phenocopies mir-92a overexpression.** If mir-92a suppresses neuronal excitability by downregulating *sirt2* expression, *sirt2* RNAi should phenocopy mir-92aOE. Three RNAi lines are available from the Transgenic RNAi Project. One shows high percentage lethality at the pupal stage at 25 °C (#36868 from Bloomington Stock Center), but the other two (#32482, RNAi-1 and #31613, RNAi-2) show ∼85% knockdown efficiency: both lower endogenous *sirt2* in heads to ∼15% when driven by *Tubulin-GAL4* (Supplementary Fig. 11).

*sirt2* was knocked down in PDF cells (PDF-GAL4; UAS-mCD8::GFP;UAS-sirt2 RNAi) with both RNAi lines, and PDF projections were maintained in the fasciculated state during the day as well as the night (Fig. 6a). At ZT2, knockdown of *sirt2* decreases maximal axonal crosses by ∼31% and ∼38%, respectively, with no significant differences observed at ZT14 (Supplementary Fig. 12). This is similar to the mir-92aOE phenotype shown above (Fig. 2a). In addition, *sirt2* RNAi abolished the effect of the mir-92aSP in maintaining the PDF

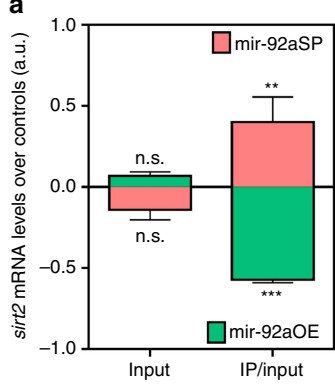

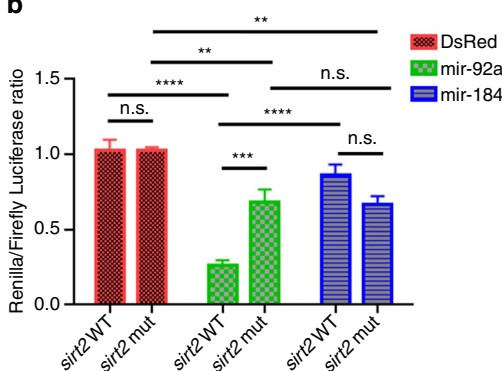

**Figure 5 | mir-92a suppresses *sirt2* translation but not mRNA levels.**
(**a**) Quantification of *sirt2* levels in TRAP. *Sirt2* mRNA levels in mir-92aOE flies were compared to those of w1118 flies, and mir-92aSP to scramble flies. *sirt2* mRNA levels in controls (w1118 and scramble) were normalized to 0. All flies expressed *Tim-GAL4;UAS-RiboTag* to tag the ribosome for immunoprecipitation. $N = 4$ (two replicates done with high-throughput sequencing and two replicates done with RT–qPCR). Error bars represent $\pm$ s.e.m., n.s. represents non-significant, \*\**P* < 0.01, \*\*\**P* < 0.001, two-way ANOVA. (**b**) Luciferase reporter assay of *sirt2* in S2 cells. *UAS*-mir-92a, *UAS*-dsRed (negative control) or *UAS*-mir-184 (negative control) was co-transfected with *Ub-GAL4* and psiCHECK2-*sirt2* or psiCHECK2-*sirt2* mut (mutant) into S2 cells for 3 days before measurement for luciferase. $N = 4$. Error bars represent $\pm$ s.e.m., n.s. represents non-significant, \*\**P* < 0.01, \*\*\**P* < 0.001, \*\*\*\**P* < 0.0001, two-way ANOVA.

projections defasciculated in both the day and the night, indicating that *sirt2* is epistatic to (downstream of) mir-92a (Supplementary Fig. 12). As *sirt2* RNAi driven by *PDF-GSG* is also sufficient to maintain projections in the fasciculated state, the phenotype is probably not due to developmental effects (Supplementary Fig. 13).

In the *in vivo* $Ca^{2+}$ imaging assay, decreased $Ca^{2+}$ levels were also observed with a *sirt2* knockdown (*PDF-GAL4;UAS-CaLexA; LexAop-luciferase;UAS-sirt2* RNAi; Fig. 6b), similar to mir-92aOE (Fig. 2c). In addition, adulthood-specific (*PDF*-GSG) sirt2 RNAi decreased PDF neuron responsiveness to nicotine stimulation (Supplementary Fig. 14), comparable to the effect of mir-92a overexpression (Supplementary Fig. 5).

In dopaminergic neurons, *sirt2* RNAi (*TH-GAL4;UAS-sirt2* RNAi) caused increased sleep, which phenocopies mir-92aOE (Fig. 6c; *sirt2* RNAi-2 increased sleep duration only when combined with *UAS-dicer2* to increase the knockdown efficiency; Supplementary Fig. 15). Knockdown of *sirt2* in flies co-expressing the mir-92aSP in dopaminergic neurons increased the sleep duration back to normal levels, indistinguishable from WT flies, indicating once again that *sir2* activity is epistatic to mir-92a

(Fig. 6d). *sirt2* RNAi also causes a bigger phase shift response at both ZT15 and ZT21 like mir-92aOE (Fig. 6e).

An amorphic strain of *sirt2* (#8839 from Bloomington Stock Center) was also assayed. We first checked whether this phenotypically null strain shows any PDF cell fasciculation phenotype. However, the projections also show substantially increased branching (Supplementary Fig. 16A), which complicates the fasciculation assay. To better quantify and distinguish defasciculation versus branch overgrowth, we assayed the defasciculation index (DI); it is the percentage of axonal intersections across concentric rings outside of a 15° cone[31]. High DI indicates defasciculation, and low suggests fasciculation. A low DI at ZT2 as well as ZT14 indicates that PDF cell projections maintain a fasciculated state in the *sirt2* amorphic strain, and increases of axonal crosses indicate an increased branching at both ZT2 and ZT14 (Supplementary Fig. 16). This indication of increased branching could be due to non-cell autonomous effects and/or a complete lack of functional SIRT2 in the animals, for example, a developmental effect.

We also assayed the *sirt2* amorphic flies behaviourally. They show $\sim 23\%$ increase in total sleep duration and almost 100% increase of sleep during LP (Supplementary Fig. 17A). This is similar to *sirt2* RNAi in dopaminergic neurons and opposite to mir-92a null flies (Fig. 3c and 6c). Strikingly, *sirt2* amorphic flies are also highly arrhythmic, showing significantly lower rhythmicity index compared to w1118 WT flies (Supplementary Fig. 17B). This may be due to the disruption of cycling neuronal excitability in circadian neurons.

In summary, *sirt2* RNAi phenocopies mir-92aOE in PDF cells and can reverse the phenotypes caused by mir-92aSP in the morphological and behavioural assays. Not surprisingly, although *sirt2* amorphic flies present a more complicated picture, the data taken together indicate that mir-92a suppresses neuronal excitability via the downregulation of *sirt2* expression.

## Discussion

Previous work on PDF cell mRNA suggested that there are many more cycling mRNAs in these pacemaker neurons than in heads[14]. Our recent RT–qPCR experiments from sorted PDF neuron RNA indicates a similar conclusion for miRNAs, namely that there are more cycling miRNAs in these pacemaker neurons than in whole heads[23]. We focus here on mir-92a, as it was the only identified miRNA under core clock control and peaking during the nighttime (Fig. 1).

Several assays showed that mir-92a suppresses neuronal excitability. They included immunostaining of PDF projections, *in vivo* $Ca^{2+}$ monitoring and imaging with ArcLight and GCaMP6 reporters during stimulation with high KCl concentrations and nicotine. All of these assays indicate that mir-92a suppresses the neuronal excitability of PDF cells (Fig. 2 and Supplementary Figs 3–6). An additional assay on sleep duration further confirms and extends this interpretation by suggesting that the suppression is not restricted to PDF cells. However, it remains possible that the PDF projection morphology effect is independent of changing neuronal excitability[39]. This could still be achieved through *sirt2*, which is able to deacetylate and destabilize microtubules and thereby affect morphology rather directly rather than only indirectly through an effect on excitability[40–42]. It is also plausible that other mir-92a targets are relevant. *Mef2* for example is a predicted target of mir-92a according to TargetScan and known to affect PDF projection morphology[31]. However, our TRAP data gave no indication that *mef2* expression is regulated by mir-92a.

miRNAs have been previously shown to regulate neuronal activity in both flies and mammals[43–45]. rno-mir-92a (rat mir-92a) is implicated in homeostatic plasticity: it is

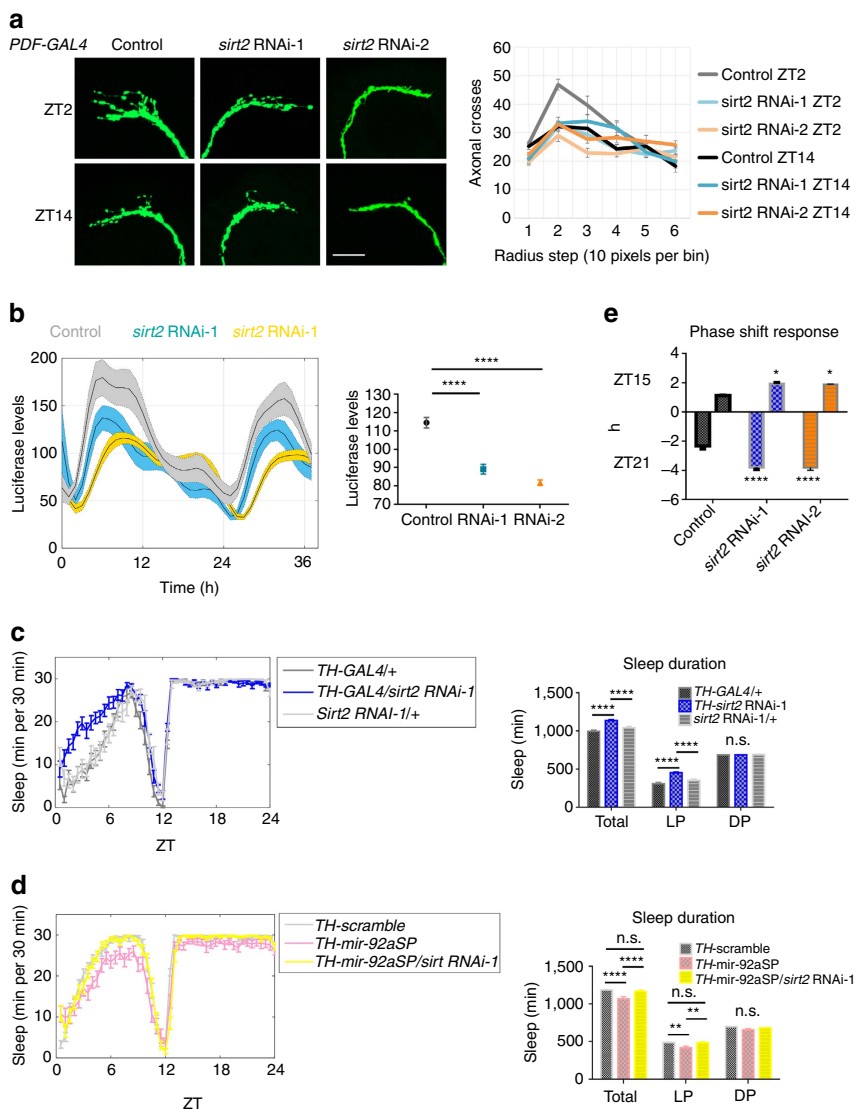

**Figure 6 | *sirt2* RNAi phenocopies mir-92aOE in PDF cells. (a)** Immunostaining of PDF cell projections with anti-GFP antibody at ZT2 or ZT14. Control is *PDF-GAL4;UAS-mCD8::GFP/+* with #36303 genetic background (flies from Bloomington Stock Center used for transgene injection). *Sirt2* RNAi are two *PDF-GAL4;UAS-mCD8::GFP;UAS-sirt2* RNAi strains (RNAi-1, RNAi-2). Left panel shows representative images of PDF cell projections of the indicated genotype at the indicated time. Scale bar, 25 μm. Quantification with Sholl analysis is shown to the right. Statistical analysis is shown in Supplementary Fig. 12. **(b)** CaLexA *in vivo* monitoring. Controls is *PDF-GAL4;UAS-CaLexA;LexAop-luciferase/+* (with #36303 genetic background). *Sirt2* RNAi are the two *PDF-GAL4;UAS-CaLexA;LexAop-luciferase/UAS-sirt2* RNAi strains. Flies were measured for luciferase levels every hour for 1.5 day at 25 °C under LD cycles. Quantified to the right is the mean of the luciferase levels. N = 12. Error bars represent ± s.e.m., ****P < 0.0001, two-tailed *t*-test. Figures shown are representative of three trails. **(c,d)** Sleep profiles of female flies entrained under LD cycles. Both *TH-GAL4/+* and *sirt2* RNAi-1/+ are controls. *TH-sirt2* RNAi-1 indicates *TH-GAL4;UAS-sirt2* RNAi-1. *TH*-scramble indicates *TH-GAL4;UAS*-scramble. *TH*-mir-92aSP indicates *TH-GAL4;UAS*-mir-92aSP. *TH*-mir-92aSP/*sirt2* RNAi-1 indicates *TH-GAL4;UAS*-mir-92aSP;*UAS-sirt2* RNAi-1. Sleep duration is quantified to the right. N = 32. Error bars represent ± s.e.m. n.s. represents non-significant, **P < 0.01, ****P < 0.0001, two-way ANOVA. **(e)** Phase shift response. Ten minutes of dim light (0.69 mW cm⁻²) exposure was given to flies of the indicated genotypes. *PDF-GAL4/+* flies were used as controls. *sirt2* RNAi-1/2 indicates *PDF-GAL4;UAS-sirt2* RNAi-1/2. Phase differences were quantified between flies subjected to light pulses or not. N = 16. Error bars represent ± s.e.m., *P < 0.05, ****P < 0.0001, two-way ANOVA. Figures shown are representative of three trails.

downresulated with TTX/AP5 treatment in cultured rat hippocampal neurons, which allows increased expression of its target *GluA1* and raises neuronal excitability[32]. The *Drosophila* mir-310 cluster has also been reported to regulate activity-dependent synaptic homeostasis, in this case by targeting *khc-73* in larval motor neurons[46]. Importantly, this cluster shares the same seed sequences as mir-92a, and rat rno-mir-92a belongs to the same family as *Drosophila* mir-92a. In both cases, these mir-92a-related miRNAs downregulate neuronal excitability, consistent with what we report here.

Similar to the response of rno-mir-92a expression to changing neuronal excitability, light pulses in the night also rapidly change PDF cell mir-92a levels. As nighttime light pulses are known to fire these cells[8], light/changing neuronal excitability as well as the core clock regulates mir-92a expression (Fig. 4a); this is the same conclusion drawn from the daily regulation of mir-92a expression (Fig. 1). Consistent with a homeostatic view, we speculate that the observed increase in mir-92a levels with a ZT15 early-night light pulse lowers PDF cell excitability and thereby helps keeps the flies longer in the 'night' state, contributing to the early-night phase

delay. However, in the late night at ZT21, mir-92a downregulation by light contributes to a more rapid increase in excitability and therefore facilitates the phase advance that occurs at this time. How light and firing have opposite effects on mir-92a levels at these two times and the mechanisms involved, for example, transcriptional or post-transcriptional regulation, are currently unknown.

These speculations about the response of mir-92a levels to light do not address the fact that overexpression or knockdown of mir-92a in PDF cells changes the magnitude of the phase shift responses (Fig. 4b). miR-92a function and neuronal excitability are therefore also upstream of the clock; this effect of mir-92a is probably indirect through its effect on firing and the effect of firing on the core clock (Fig. 7)[8]. It has been previously reported that mice with lower neuronal excitability in the suprachiasmatic nucleus show bigger phase delays with an early-night light pulse. The authors reasoned that this may be because of a more labile clock due to a higher sensitivity to environmental stimulus when neuronal excitability is suppressed[47]. This is also similar to classical limit cycle theory, that is, an oscillator with reduced amplitude shows enhanced phase shifts[48]. Flies similarly show bigger phase shifts with lower neuronal excitability (mir-92aOE; Fig. 4b), suggesting some commonality in mechanism.

We identified *sirt2* mRNA as a mir-92a target responsible for many of the observed phenotypes (Fig. 6). Its regulation was shown by TRAP and by a S2 cell reporter assay (Fig. 5). *Sirt2* is an attractive target as it has been reported (1) to reduce neuronal branching in an RNAi screen in fly sensory neurons[49]; (2) to influence neuronal metabolism in mammals and flies by regulating gene expression through acetylation of histones[40]; (3) to regulate mitochondrial energy metabolism through acetylation of mitochondrial complex V (ref. 50); and 4) to regulate neuronal excitability in mice: the inhibitor of SIRTUINS decreases neuronal excitability and the activator increases it[51]. Although it is unclear which of these pathways are upstream of the observed phenotypes, we favour at a minimum a role for neuronal excitability.

Since *sirt2* mRNA does not cycle in PDF cells[52], the purpose of cycling mir-92a levels may be to generate cycling SIRT2 levels; high levels in the morning would contribute to higher neuronal

excitability at this time. We also expect changes in PDF cell SIRT2 levels with light pulses. However, western blots and immunostaining experiments are missing because of the lack of specific anti-SIRT2 antibodies.

Taking all the results into consideration, we suggest that mir-92a expression is under the control of light and the core molecular clock in PDF cells. Cycling mir-92a levels target *sirt2* by imposing higher levels of suppression at night and lower in the morning. This generates cycling levels of SIRT2, which enhance neuronal excitability and downstream events in the daytime. They include PDF projection morphology and behavioural consequences such as sleep as well as the phase shift response (Fig. 7).

## Methods

***Drosophila* stocks.** Flies were reared on standard cornmeal/agar medium with yeast under 12:12 h LD cycles at 25 °C. *PDF-GAL4;UAS-mCD8::GFP* was described in ref. 53. *UAS-mir-92aOE, PDFR-GAL4* and *UAS-sirt2* RNAi flies were from the Bloomington stock centre. *UAS-mir-92aOE* was backcrossed six times to w1118 flies. *UAS-scramble* and *UAS-mir-92aSP* were kind gifts from Davie Van Vactor lab. *UAS-ArcLight* flies were described in ref. 11. *UAS-GCaMP6f* was described in ref. 54. *PDF-GAL;UAS-CaLexA;LexAop-luciferase* was a recombined stable line as described in ref. 34. *PDF-GSG* was described in ref. 30. *TH-GAL4* was described in ref. 8. *PDF-GAL4* was described in ref. 9. *Dvpdf-GAL4* was a kind gift from Dr JH Park. *UAS-RiboTag* was a gift from the Zipursky lab.

**Plasmids.** psiCHECK2 plasmid is commercially available from Promega. To insert the *sirt2* 3′UTR, the vector was first digested with XhoI and NotI and the PCR product amplified from gDNA with the *sirt2* 3′UTR forward and reverse primers (Supplementary Table 2) and then incorporated into the vector with Gibson Assembly (NEB). To mutate the mir-92a-binding site in the *sirt2* 3′UTR, the Agilent Quickchange Kit was used with the following primers: sirt2 mut 3′UTR forward and reverse (Supplementary Table 2).

**Sleep and phase shift assay.** Trikinietics Acitivity Monitors (Waltham, MA) were used to measure the locomotor activity of individual flies around 7-day old. For sleep assays, female flies were used, and the data analysed with sleep analysis scripts developed by the Griffith lab at Brandeis University using MATLAB (MathWorks, Natick, MA). For phase shift assays, male flies were entrained and subjected to a 10-min 0.69 mW cm$^{-2}$ light pulse at a given time point and left in DD for 7 days. Circadian rhythm behaviour including rhythmicity, period and phase shift was analysed as described[8].

**Fly brain immunocytochemistry.** Immunostaining was done as described[38]. Briefly, fly heads were fixed in PBS with 4% paraformaldehyde supplemented with

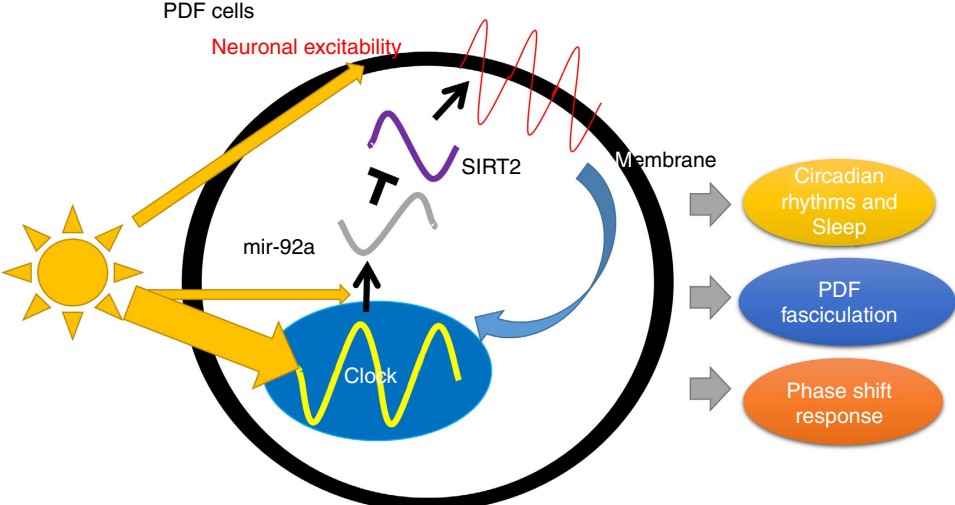

**Figure 7 | Model.** The circadian core clock in PDF cells regulates daily cycling expression of mir-92a. In addition, mir-92a expression levels are regulated by light either directly by unknown mechanisms or indirectly through effects of light on the core clock. mir-92a suppresses neuronal excitability by inhibiting translation of *sirt2*, leading to downstream changes, such as daily oscillation of PDF projection fasciculation, phase shift responses and sleep (which is affected by circadian rhythms at most times). Light also affects the neuronal excitability of PDF cells, which changes mir-92a levels possibly through the core clock[8].

0.008% Triton X-100 for 1 h at 4 °C before dissection. A mouse anti-GFP antibody (1:1,000, Sigma-Aldrich, G6539), a mouse anti-PDF antibody (1:10, Developmental Studies Hybridoma Bank, University of Iowa, Iowa city, IA, described in ref. 8) and Alexa Fluor 488 (1:500, Invitrogen Cat#: A-11001) were used as primary and secondary antibodies. Brains were imaged at × 20 on a Leica SP5 confocal microscope. Images are maximum projections of Z sections. Sholl analysis in FIJI was used for axonal crosses, and quantification was according to software instructions. DI was calculated with modified Sholl analysis as described in ref. 31.

**ArcLight and GCaMP6f imaging.** Fly brains were dissected in AHL consisting 108 mM NaCl, 5 mM KCl, 2 mM CaCl$_2$, 8.2 mM MgCl$_2$, 4 mM NaHCO$_3$, 1 mM NaH$_2$PO$_4$, 5 mM trehalose, 10 mM sucrose and 5 mM HEPES, and mobilized to the bottom of a perfusion Sylard-bottom (Dow Corning, Midland, MI) chamber filled with AHL using a pin anchored to the Sylgard[55,56]. Depolarization buffer (high KCl for ArcLight experiments) containing 28 mM NaCl, 85 mM KCl, 2 mM CaCl$_2$, 8.2 mM MgCl$_2$, 4 mM NaHCO$_3$, 1 mM NaH$_2$PO$_4$, 5 mM trehalose, 10 mM sucrose and 5 mM HEPES, or $3 \times 10^{-6}$ M Nicotine diluted in AHL (for GCaMP6f experiments, Sigma-Aldrich N3876, concentration modified according to ref. 33) was perfused into the chamber using a gravity-fed ValveLink perfusion system (Automate Scientific, Berkeley, CA)[56,57]. Imaging was done with an Olympus BX51WI fluorescence microscope (Olympus, Center Valley, PA) under an Olympus × 60 (0.90 W, LUMPlanFI) water-immersion objective and was captured using a charge-coupled device camera (Hamamatsu ORCA C472-80-12AG). The following filter sets were used for excitation and emission (Chroma Technology, Bellows Falls, VT): excitation, HQ470/× 40; dichroic, Q495LP; emission, HQ525/50 m. Frames were captured with μManager with 2 Hz with 4 × 4 binning with 500 ms exposure time and 50 ms intervals[58]. Fluorescence levels were quantified with FIJI.

**CaLexA bioluminescence recording.** The recording was done as described in ref. 34. Basically, food containing 1% agar and 5% sucrose was heated to melt and supplemented with 20 mM of D-luciferin potassium salt (GOLDBIO), 250 μl of which was then distributed into every other well in a 96-well plate. Plates were allowed to cool down completely before use. Individual flies were loaded into each well and the plate was then sealed with a transparent adhesive (TopSeal-A PLUS, Perkin Elmer). Every well was punctured with two to three small holes to allow air circulation, and recording was with a TopCount NXT luminescence counter (Perkin Elmer) in an incubator under LD cycles at 25 °C. Data were then analysed with MATLAB.

**RNA extraction and qRT-PCR.** For cell-extracted RNA, ∼ 100 GFP-labelled PDF cells were manually sorted and stored in 100 μl TRIzol reagent (Invitrogen). A detailed procedure for cell sorting is described in ref. 28. RNA was then extracted following the supplier's protocol. To quantify individual miRNAs, total extracted RNA was ligated with a 3′ adaptor and then a 5′ adaptor for RT–PCR. The PCR product was then diluted (1:20) and quantified by qPCR using a universal reverse primer and a miRNA-specific forward primer (Supplementary Table 2). 2S rRNA was amplified along with miRNAs and served as a normalization RNA. The strategy is adapted from a miRNA deep-sequencing protocol and allows easier and more higher-throughput screening. Stem–loop qPCR confirmed the results for mir-92a (Supplementary Fig. 18).

**TRAP.** Flies expressing *Tim-GAL4;UAS-RiboTag* (FLAG tag) in addition to mir-92a manipulation (*UAS*-mir-92aOE (w1118 as control) or *UAS*-mir-92aSP (*UAS*-scramble as control)) were collected on dry ice and decapitated. Fly heads were homogenized (lysate kept as input) and immunoprecipitated with Sigma M2 anti-FLAG magnetic beads. RNA was then extracted from the beads with TRIzol reagents (IP). RT–qPCR and high-throughput sequencing were performed with both the Input and IP RNA for quantification. For high-throughput sequencing, data were mapped to dm3 genome using Tophat[59] and expression levels were quantified using Cufflinks[60].

**S2 cell luciferase assay.** S2 cells plated in 96-well plates (Costar, 3610) were co-transfected with 12.5 ng of each plasmid mixed with 2 μl of Cellfectin II Reagent (Thermo Fisher). Plasmids included *Ub-GAL4, UAS*-mir-92a, *UAS*-dsRed, *UAS*-mir-184 and psiCHECK2. The Promega Dual-Luciferase Reporter Assay System was used to measure luciferase levels 3 days post transfection. *Renilla* is the reporter for 3′UTR activity, and *luciferase* is the transfection efficiency control (Supplementary Fig. 10).

**Data availability.** The authors declare that all data supporting the findings of this study are available within the article and its Supplementary Information, or from the corresponding author on request.

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

## Acknowledgements

We thank members of Rosbash lab for thoughtful discussion, especially Kate Abruzzi for comments and Fang Guo for sharing the newly developed CaLexA system. We also thank members of Griffith lab for help with the ArcLight/GCaMP6 imaging, the perfusion system and sleep analysis. We thank Drs Orie Shafer and Patrick Emery for helpful discussions as well as Dr Stephen Cohen for freely sharing mir-92aKO flies.

## Author contributions

X.C. and M.R. designed the experiments and wrote the manuscript. X.C. performed the experiments.

## Additional information

**Competing financial interests:** The authors declare no competing financial interests.

