## [Peer Review File · Nature Communications]

Reviewers' comments:

Reviewer #1 (Remarks to the Author):

The manuscript from Chen and Rosbash identifies a specific microRNA (mir-92a) that is rhythmically expressed in the master pacemaker clock neurons of *Drosophila*. They propose that mir-92a alters the excitability of clock neurons and this effect is mediated by regulation of Sirt2. The topic is interesting and the identification of miRNAs in a specific cell type is impressive, but there are also major weaknesses that need addressing.

Major comments.

1. I am not convinced about the phase advance in DD of mir-92a expression. It is clear for day 1 in figure 1b, but not clear for the day2 in figure 1b. It looks better in figure S2, but I do not think these are the same data as figure 1 since day 1 in LD looks the same as day 2 and suggests that the data are double-plotted. Is that correct?

2. If mir-92a expression is regulated by light, then it should show different levels in LD cycles in per mutants (figure 1c) like Tim protein. This graph should also have light and dark bars like figure 1a. The point about regulation by light is even more confusing with the effects on expression in the phase shift experiments.

3. One major absence is any discussion on circadian behavior of the OE or sponge lines. Since previous work from the Nitabach lab and others has shown that altering PDF cell excitability can change period and alter rhythmicity, does altering mir-92a expression affect period length? Failing to mention this suggests that the authors found no phenotypes, but this should at least be discussed.

4. The labeling and presentation of the graphs needs improving. There are many examples of this, including:

CT26 is standard in the field, but I have never seen ZT26 (figure 1a) before.

Figure S2 has ZT on the x-axis but refers to LD and DD cycles.

The numbers on the x-axis in figure 2b are unhelpful (41.8, 83.05 etc) and overlap with numbers on the y-axis.

Why are the fonts different sizes in 2C top and bottom?

5. In figure 2b, why are the Arlight fluorescent levels so different between the control and scramble in the top and bottom graphs? OE, scramble and sponge all look similar, while the control is the one that looks different. This makes me wonder if the effects are non-specific and due to expression of anything in PDF cells. What is the genotype of the control?

6. Why are there now two peaks now with the sponge in figure 2c?

7. Could mir92 suppress excitability indirectly by affecting the projections?

8. Have the authors tested if the effects of expressing mir-92a are developmental or adult-specific?

9. The homologous function for rat mir92a in regulating excitability is mentioned in the discussion but must be mentioned much earlier. Otherwise it looks like the authors are trying to take the credit for this idea.

Minor comments.

1. The significance statement should be broadened and should read differently from the abstract.

2. In the introduction to the circadian system: is it "150 pairs of neurons" or just "150 neurons"? I think the latter.
3. The authors should explain how they chose to quantify the 16 specific miRNAs? "Based on hints from the sequencing" is too vague.
4. For figure 2, the "maximal axonal crosses at ZT2" needs explaining. This is not intuitive to the non-expert.
5. Please explain that CaLexA is an artificial transcription factor.
6. Please introduce sirt2.
7. Please use standard formal scientific prose. For figure 6, "of which one is sick" is too vague and the reader has no idea if this is the RNAi line itself or the cross in which the RNAi is expressed in specific cells.
8. The figure legends need attention. For example, figure 2a "was done in" is poor English, but this whole sentence is difficult to read. This kind of comment that I am making is not a good reason to reject a paper, but it creates a barrier through which the reader has to penetrate to understand the experiment and this happened a lot in this paper.
9. There is far too much data crammed into figure 6 and it is hard to see.

Reviewer #2 (Remarks to the Author):

This manuscript provides evidence supporting the conclusion that circadian pacemaker neuron and sleep-regulating dopamine neuron electrical excitability is regulated by mir-92a acting through sirt2. This is an interesting conclusion and the experiments as presented mostly provide robust results. However, there are a few key experiments necessary to convincingly support the overall conclusions of the paper.

Specifically, the following points need to be addressed:

- (1) Does the mir-92a precursor transcript have E-boxes or other circadian regulatory elements?
- (2) The Arlight experiments are unconvincing and uninformative as performed using KCl-mediated depolarization, and the authors should image at high-speed as in the Cao et al., 2013 paper in Cell to directly assess spontaneous electrical activity in the sLN_v dorsal terminals.
- (3) sirt2 RNAi experiments are not properly controlled, and require as a negative control expression of a different RNAi line targeting a transcript not expressed in the cells of interest. This is necessary to rule out sirt2-independent off-target non-specific effects of RNAi expression.
- (4) To directly assess sirt2 effects on neuronal excitability, Arlight experiments directly assessing spontaneous neuronal activity as described above should be performed on sirt2 RNAi flies (with appropriate negative controls) and on sirt2 overexpression flies.

Reviewer #3 (Remarks to the Author):

The authors study the dynamics, regulation about potential function of cycling miRNAs in sleep promoting and sleep inhibiting neural centers of the Drosophila brain. They use a host of sophisticated genetic and behavioral methods to argue that mir-92a is a cycling regulator of target RNAs in critical circadian pacemaker cells, called PDF neurons. They accumulate impressive data to conclude that this mir inhibits neuronal excitability in PDF cells at certain times. Also, by completing an exhaustive target search, they propose that mir-92a actions are primarily via its inhibition of sirt2. All in all this is a very

impressive effort which sheds much new light the physiology of pacemaker cells types and interprets its results conservatively. With minor exceptions (noted below, the Figures are clear and statistics appropriate. I have a few comments and questions that may improve the manuscript.

My primary concern was the application of chronic genetic manipulations and the overall lack of any conditional genetic designs. In general fly genetics is turning increasingly to the use of conditional designs to avoid the confusion between impairing developmental properties versus mature ones. In the case of these neurons, there is clear precedent that chronic knock down by RNA can have deleterious effects on cell viability, or can affect both developmental and mature cell properties. There are a lot of experiments reported in this submission, so repeating all experiments is not a reasonable request. However, confidence could be greatly increased by performing one or two of the key experiments in which mir92a levels are up and down regulated (measured with for example sleep assays, or ARC light assays) with a conditional design.

Minor points

Figure 2C. With expression of the SP isoform of 92a, calcium levels in PDF cells in vivo displayed not just elevated levels, but two distinct peaks in each of three consecutive cycles. This was an interesting finding – but there was no specific comment.

Figure S2. I was confused by this Figure in relation to Figure 1 of the Main text. Are these both performed with purified PDF cells? It looks different from the data in Figure 1, but I am not sure why it does. Fig S2 shows a consistent phase difference across two cycles, but in Figure 1 it is only apparent in the first cycle. Also, 2 biological replicates were superimposed – does that mean each was 48 hours? Or were they 24 hr each and concatenated?

Figure S4. Were movement artifacts observed with Hi K? A more traditional graphic feature is to use a bar to indicate the time period when Hi K was perfused (don't just describe it in the figure legend). Was there a difference between expressing one versus two copies of UAS-kir? On a more general note, I was confused by increased kir mitigating the depolarization effects of Hi potassium (line 148-150) - an explanation would be helpful. I thought the membrane potential was essentially controlled by E[Cl] with hi K, so why should additional inward rectifying K channel change that? Also a technical description of the kir transgene is needed - I found none.

Figure S6. Were values between ZT16 and ZT22 significantly different?

Figure S 9. I was confused by discussion of the sirt2 RNAi stocks – the text (lines 249-252) describes on sick and one ineffective but shows two others; would be cleared up by simply using numerical identifiers for each of the three. Also – why was this data normalized to Rpl32, while other RT-PCR reported (like Figure S6) was normalized 2s rRNA?

Figure S10. Axonal crossing are quantified but there is no indication of statistical analysis.

Figure S11. Error bars are shown but N=2.

Table S1. Suggest reporting raw values and not the average of two biological replicates.

Reviewer #4 (Remarks to the Author):

In their manuscript "MicroRNA-92a is a Circadian Modulator of Neuronal Excitability in Drosophila",

Rosbash and colleagues characterize mir-92a as a cycling locus in PDF cells, a key population of circadian pacemakers. They utilize a battery of functional assays, including behavioral activity profiling and neuronal activity measurements, to show that mir-92a suppresses neuronal excitability. Finally, they assign sirt2 as a functionally relevant target of mir-92a that appears to mediate some of the phenotypes in mir-92a depletion conditions.

I did enjoy reading the paper, and overall I support it for publication once it is appropriately revised. They have used an impressive diversity of sophisticated neurobiological and molecular techniques, which puts this study at the fore of investigations on in vivo miRNA function. The paper is overall easy to follow. However, I have some experimental comments regarding experiments that should be addressed.

1. I find it odd they never use the available KO mutants that were generated by multiple labs and are available at the Bloomington Stock Center. It is possible that null flies do not necessarily have to show the same phenotypes as the tissue specific effects reported, since other behaviors may be masking or even making impossible to carry out the assays, but at least they should try it. Also this can help address the contribution of the related mir-92a and mir-92b, since different papers have addressed overlapping in vivo requirements for these miRNAs in Drosophila (eg. Yuva-Aydemir Plos Genetics 2015), and a sponge cannot distinguish these isoforms. The above paper indicates that mir-92a and mir-92b are cotranscribed, and thus one may infer they both have rhythmic expression and affect some processes they studied. (For that matter, since these miRNAs are also produced from the jigr1 gene, one also wonders whether jigr1 is on the list of cycling loci. Or whether it has a function in rhythmic behavior, although I believe this is outside the scope of this study).

In any case, I consider it necessary to confirm at least some of the key behavioral phenotypes and/or anatomical results on PDF cell projections attributed from sponge expression using mutants.

2. Since the story begins with studying cycling miRNAs in PDF cells, the core circadian pacemakers, it is odd why there don't seem to be any data regarding rhythmicity when the miRNA is manipulated. This seems like very basic data that they probably generated, but if the results were either positive or negative they should describe the data.

3. The manuscript also rises an interesting consideration: do cycling miRNA levels correspond to cycling miRNA activity? The results and paper are written to suggest so. However, the data are also compatible with developmental defects. Also, previous studies on mir-92a/b have indicated developmental defects in the nervous system. It would be interesting to see if the effects observed for OE and SP are reversible if GAL4 expression ceases. This could give some fruitful information on the correlation of miRNA levels oscillation and miRNA activity oscillation.

For example, the effects from mir-92a and mir-92a-SP are expressed constitutively in PDF cells, and not with temporal control. This could imply that the effects observed in PDF termini are a developmental aberration induced by the manipulation of miRNA levels, rather than an adult specific phenotype, and thus that mir-92a cycling has no consequence on PDF cells. If the hypothesis of the authors is that distinct levels of mir-92a induce different PDF termini conformations, the authors should perform these manipulations strictly in the adult stage, and show its reversibility. I suggest that they add tubGal80ts to the genotype of the OE and SP flies, and show that while raised at 18C, flies show the expected PDF termini at ZT2 and ZT14, but when transferred to 29C, both ZT look alike (fasciculated for OE and defasciculated for SP). Then, upon transferring the flies back to 18C, the PDF termini cycle normally again.

4. Sirt2. There are published sirt2 mutant flies and even sirt2 gain of function flies. Similar to the mir-92a studies, the role of sirt2 is not tested with genuine mutants, which are available at the Bloomington Stock Center. I would like to see this confirmed.

Minor comments:

miRNA quantification:

.- Fig. 1A: miRNA profile in the two light periods, and in the two dark phases shown seem very different in terms of levels, peaks, and tendency. Also, in fig 1C, in the absence of per protein, miRNA levels amplitude of variation is as high as observed under wt conditions in Fig.1A. Why are the authors convinced then, that fig.1A is unambiguously showing cycling of miRNA levels, and not just showing some unspecific variation? Also, the mir-92a levels in Fig.1A and in Fig. S2 look quite different, with the latter showing almost an idealized regular up and down pattern, much more regular than the former. Showing the expression range from the replicate data would be useful to interpret.

Neuronal excitability:

.- The genotype of the control is not stated for the OE expression in fig 2B, while it is stated in fig 1A. The controls here should be the Gal4+arc light and UAS+ArcLight in both experiments.

.- What is the ZT of the brains used in the ArcLight experiment? Since the time of day is determinant for neuronal excitability, I assume that all the brains were synchronized, although this is not specified in the manuscript. This should be clarified. If I understand, for the OE data, both control and experimental should be ZT2 (max excitability in the controls and reduced in the OE). For the SP experiment, it should be done in ZT14 (min excitability for the control, increased in the experimental). Is this the design used? If another design was used, then the authors should clarify.

CaLexA:

.- Line 164: "especially during the morning". I am not sure if I interpreted right, but it seems the main reduction is more apparent in the second half of the light phase, that would correspond to the afternoon or evening, but in any case the morning (Fig2. C OE panel).

.- mir-92a SP levels show an acute peak at night. Nevertheless, if neuronal excitability depends on miR-92a levels (and correlate to PDF conformations), and these are kept uniform in the KD genotype, shouldn't the excitability be constant as well? Why does it clearly decays in the beginning of the night and during the morning instead of showing a plateau? Same for OE experiments.

Sirt2:

.- Line 230: They say "mutant flies" and that is not correct, they mean KD

.- The conclusion raised in the last sentence is overstated. From the in vitro sensor experiment, they can only state that mir-92a suppresses sirt2 expression in vitro by binding to the mentioned site. They cannot be sure if it does so in vivo, for that they need to delete that binding sites in live animals and check sirt2 expression in those conditions.

RESPONSE TO REFEREES

Reviewers' comments:

Reviewer #1 (Remarks to the Author):

The manuscript from Chen and Rosbash identifies a specific microRNA (mir-92a) that is rhythmically expressed in the master pacemaker clock neurons of *Drosophila*. They propose that mir-92a alters the excitability of clock neurons and this effect is mediated by regulation of Sirt2. The topic is interesting and the identification of miRNAs in a specific cell type is impressive, but there are also major weaknesses that need addressing.

Major comments.

1. I am not convinced about the phase advance in DD of mir-92a expression. It is clear for day 1 in figure 1b, but not clear for the day2 in figure 1b. It looks better in figure S2, but I do not think these are the same data as figure 1 since day 1 in LD looks the same as day 2 and suggests that the data are double-plotted. Is that correct?

Your confusion is justified as we should have been more specific. The data in Fig. S2 was the average of the two replicates, which are indeed double plotted. This is now specified. In LD, mir-92a peaks at ZT22, but the data are more variable in DD: one replicate shows a 8 hr advance (peaks at CT14) whereas the other shows a 4 hr advance (peaks at CT18); hence the claim of a 4 – 8 hr advance. We have now completed and added a 3rd replicate of Fig.1, where DD once again shows 4 hour phase advance compared to LD. With 2 replicates of a 4 hr advance and 1 of an 8 hr advance, we are confident that there is an advance even if the magnitude is uncertain.

2. If mir-92a expression is regulated by light, then it should show different levels in LD cycles in per mutants (figure 1c) like Tim protein. This graph should also have light and dark bars like figure 1a. The point about regulation by light is even more confusing with the effects on expression in the phase shift experiments.

The minor point about the background light-dark bars has been corrected. As for the more important point about light regulation, we suspect the core clock may have a major role in regulating mir-92a expression and the light has a more minor role; it is even possible that light acts through the core clock. Therefore, without a functional core clock, mir-92a expression fails to show substantial light-dependent expression under LD conditions. We do not understand the TIM analogy. Perhaps the reviewer expects a qualitative effect of light like on TIM (near total disappearance), but the effect on mir-92a could be much more subtle.

3. One major absence is any discussion on circadian behavior of the OE or sponge lines. Since previous work from the Nitabach lab and others has shown that altering PDF cell excitability can change period and alter rhythmicity, does altering mir-92a expression affect period length? Failing to mention this suggests that the authors found no phenotypes, but this should at least be discussed.

Fig. S9 as well as additional text has been added under the section 'Light pulses at night alter mir-92a levels in PDF cells, and mir-92a levels affect the phase shift response' to address this issue. We note here that we expect the effect of miR-92a on excitability to be much more modest than what is in the previous literature.

4. The labeling and presentation of the graphs needs improving. There are many examples of this, including: CT26 is standard in the field, but I have never seen ZT26 (figure 1a) before. Figure S2 has ZT on the x-axis but refers to LD and DD cycles. The numbers on the x-axis in figure 2b are unhelpful (41.8, 83.05 etc) and overlap with numbers on the y-axis. Why are the fonts different sizes in 2C top and bottom?

Thank you . This is now fixed.

5. In figure 2b, why are the Arclight fluorescent levels so different between the control and scramble in the top and bottom graphs? OE, scramble and sponge all look similar, while the control is the one that looks different. This makes me wonder if the effects are non-specific and due to expression of anything in PDF cells. What is the genotype of the control?

The two graphs were done separately. It is our experience, everyone in our lab and in the neighboring Griffith lab at Brandeis, that each trial has substantial variation due to different basal fluorescence intensity, different buffers, frankly who knows what?. So each experiment must always be done side-by-side with the controls. In addition, what is shown is representative of 3 trials, each of which gave the same qualitative result without exception.

The genotype for the 'control' is *Pdf-GAL4;UAS-arclight/+* with w1118 background is indicated in the figure legend. W1118 is also the genetic background of *UAS-mir-92aOE* (backcrossed 6 times) as now specified.

6. Why are there now two peaks now with the sponge in figure 2c?

We don't know exactly why. Our hypothesis is mir-92aSP has a bigger effect on the luciferase level when mir-92a expression in PDF cells are high at night.

7. Could mir92 suppress excitability indirectly by affecting the projections?

This is possible but unlikely in our view. Manipulation of projection morphology has never been shown to affect excitability, whereas the reverse gives a clean result, i.e., firing PDF neurons with TrpA1 is sufficient to produce strong and rapid defasciculation of the projections (Sivachenko et al., 2013).

8. Have the authors tested if the effects of expressing mir-92a are developmental or adult-specific?

We added important experiments using PDF-GSG (geneswitch, feed drug only during adulthood) for projection morphology and *gcamp6* in response to nicotine. Both of these approaches indicate that the observed phenotypes are not (exclusively) due to developmental defects (Fig. S4, S5, S14, S15). Phase shift experiments were also performed with PDF-GSG. Although similar changes were observed, they were not significant probably due to the fact that PDF-GSG is much weaker than PDF-gal4. (This is very obvious when the two drivers were crossed with GCaMP6. PDF-GAL4 shows much brighter fluorescence signals than PDF-GSG).

9. The homologous function for rat mir92a in regulating excitability is mentioned in the discussion but must be mentioned much earlier. Otherwise it looks like the authors are trying to take the credit for this idea.

We agree that this was a serious (but unintended) oversight, which we have now corrected. FYI, we have been working on this problem for more than 3 years, and the rat paper appeared when we already had enough of this story completed to connect our fly 92a with excitability. Along with the completely different target, which we also identified quite some time ago, we just lost track of the fact that the rat paper was precedent-setting. Our fault.

Minor comments.

1. The significance statement should be broadened and should read differently from the abstract.

Revised accordingly.

2. In the introduction to the circadian system: is it "150 pairs of neurons" or just "150 neurons"? I think the latter.

Sorry for the mistake. It is 150 neurons. It is corrected.

3. The authors should explain how they chose to quantify the 16 specific miRNAs? “Based on hints from the sequencing” is too vague.

The deep sequencing showed some hint of cycling of these miRNAs, some of which the Qpcr showed similar results but some don't. It is now revised accordingly.

4. For figure 2, the “maximal axonal crosses at ZT2” needs explaining. This is not intuitive to the non-expert.

Revised accordingly.

5. Please explain that CaLexA is an artificial transcription factor.

Revised accordingly.

6. Please introduce sirt2.

Revised accordingly.

7. Please use standard formal scientific prose. For figure 6, “of which one is sick” is too vague and the reader has no idea if this is the RNAi line itself or the cross in which the RNAi is expressed in specific cells.

Revised accordingly.

8. The figure legends need attention. For example, figure 2a “was done in” is poor English, but this whole sentence is difficult to read. This kind of comment that I am making is not a good reason to reject a paper, but it creates a barrier through which the reader has to penetrate to understand the experiment and this happened a lot in this paper.

Revised accordingly.

9. There is far too much data crammed into figure 6 and it is hard to see.

Revised accordingly.

Reviewer #2 (Remarks to the Author):

This manuscript provides evidence supporting the conclusion that circadian pacemaker neuron and sleep-regulating dopamine neuron electrical excitability is regulated by mir-92a acting through sirt2. This is an interesting conclusion and the experiments as presented mostly provide robust results. However, there are a few key experiments necessary to convincingly support the overall conclusions of the paper.

Specifically, the following points need to be addressed:

(1) Does the mir-92a precursor transcript have E-boxes or other circadian regulatory elements?

mir-92a is embedded in the intron of the protein-coding gene *jigr1*, and there are several canonical E-boxes in the promoter region of that gene. However, *jigr1* doesn't show cycling expression in PDF cells according to our unpublished high throughput sequencing data (just submitted). So either pri-mir92a transcription is regulated

separately or the regulation is post-transcriptional. However, the expression of pre-miRNA and pri-miRNA in PDF cells is too low to be detected by qPCR.

(2) The Arlight experiments are unconvincing and uninformative as performed using KCl-mediated depolarization, and the authors should image at high-speed as in the Cao et al., 2013 paper in Cell to directly assess spontaneous electrical activity in the sLNv dorsal terminals.

We tried very hard to repeat what was done in Cao et al., but failed to observe spontaneous electrical activity; only noise (signal fluctuation that's insensitive to TTX treatment). This may be due to our microscope or light source. Orié Shafer, who is on sabbatical in our lab this year, said he too has had trouble with this assay as has the Griffith lab. Nonetheless, we tried hard to establish a local connection on another microscope but with no success. If the reviewers find this experiment unconvincing, we can remove the figure – given how data-intensive our paper is. As an alternative, GCaMP6 was used for observing spontaneous activity (data not shown, Fig. S6). GCaMP6 was also used as another approach to monitor Ca²⁺ levels in PDF cells. Flies expressed UAS-GCaMP6f together with either UAS-mir-92aOE or UAS-mir-92aSP driven by PDF-GAL4 were dissected and fluorescence levels of their PDF termini measured and quantified. While it is hard to accurately estimate spiking rates and amplitudes from the spontaneous Ca²⁺ signals due to noise and heterogeneity within individual neurons as well as among biological replicates, the differences between baseline fluorescence levels of the different genotypes were obvious. Consistent with the in vivo CaLexA results, knocking-down mir-92a results in a ~224% increase in baseline fluorescence levels at ZT18 – 22, and overexpression results in a ~40.6% decrease at ZT6 – 10 (Fig. S6). We also added experiments similar to KCl stimulation but with nicotine which is a more physiological agonist (Fig S5). Although not statistically significant, probably because of large variation, the trends are apparent and similar to the responses to KCl.

(3) *sirt2* RNAi experiments are not properly controlled, and require as a negative control expression of a different RNAi line targeting a transcript not expressed in the cells of interest. This is necessary to rule out *sirt2*-independent off-target non-specific effects of RNAi expression.

The background line used in the paper is suggested by Bloomington Stock Center as the best control lines to use for experiments. However, we have many negative controls as we did RNAi screening to find *sirt2*. Since it may be bulky to put them in the manuscript, we attach for the reviewers and editor the figure below. If required, we are happy to add it as a supplemental figure.

(Left) CaLexA monitoring with an additional control RNAi line against stat92E. Quantification at the bottom.

(Right) Sleep profiles with additional control RNAi lines against CG3678, CG3534, CG11807, CG3534 respectively. All RNAi are driven by *TH-GAL4/UAS-dicer2*. Quantification at the bottom.

(4) To directly assess *sirt2* effects on neuronal excitability, ArcLight experiments directly assessing spontaneous neuronal activity as described above should be performed on *sirt2* RNAi flies (with appropriate negative controls) and on *sirt2* overexpression flies.

This is the same as question #2 of reviewer #2. The nicotine stimulation experiments were performed, which indicate that the *sirt2* RNAi neurons are less excitable than the controls (Fig. S14).

Reviewer #3 (Remarks to the Author):

The authors study the dynamics, regulation about potential function of cycling miRNAs in sleep promoting and sleep inhibiting neural centers of the *Drosophila* brain. They use a host of sophisticated genetic and behavioral methods to argue that mir-92a is a cycling regulator of target RNAs in critical circadian pacemaker cells, called PDF neurons. They accumulate impressive data to conclude that this mir inhibits neuronal excitability in PDF cells at certain times. Also, by completing an exhaustive target search, they propose that mir-92a actions are primarily via its inhibition of *sirt2*. All in all this is a very impressive effort which sheds much new light on the physiology of pacemaker cell types and interprets its results conservatively. With minor exceptions (noted below, the Figures are clear and statistics appropriate. I have a few comments and questions that may improve the manuscript.

My primary concern was the application of chronic genetic manipulations and the overall lack of any conditional genetic designs. In general fly genetics is turning increasingly to the use of conditional designs to avoid the confusion between impairing developmental properties versus mature ones. In the case of these neurons, there is clear precedent that chronic knock down by RNA can have deleterious effects on cell viability, or can affect both developmental and mature cell properties. There are a lot of experiments reported in this submission, so repeating all experiments is not a reasonable request. However, confidence could be greatly increased by performing one or two of the key experiments in which mir92a levels are up and down regulated (measured with for example sleep assays, or ARC light assays) with a conditional design.

We agree with this point as described above to reviewer #1 in response to his/her point 8. For sleep assays, there are unfortunately no geneswitches available yet. For technical, background and temperature issues, we could not use *tub-gal80ts* with the 3 month time limit. However, we added experiments using PDF-GSG (geneswitch, feed drug only during adulthood) for projection morphology assays and *gcamp6* imaging in response to nicotine. Both assays suggested that the phenotypes observed are not solely due to developmental defects. As mentioned above, phase shifts experiments were also performed with PDF-GSG. Although similar changes to PDF-gal4 were observed, the changes were insignificant probably due to the fact that PDF-GSG is much weaker than PDF-gal4. (This is very obvious when the two drivers were crossed with *GCaMP6*. PDF-GAL4 shows much brighter fluorescence levels than PDF-GSG).

Minor points

Figure 2C. With expression of the SP isoform of 92a, calcium levels in PDF cells in vivo displayed not just elevated levels, but two distinct peaks in each of three consecutive cycles. This was an interesting finding – but there was no specific comment.

We are not sure about the exact mechanism for this observation. The only explanation we could come up with is the mir-92aSP has a bigger effect when the endogenous mir-92a is at its peak level. We have added this interpretation to the manuscript.

Figure S2. I was confused by this Figure in relation to Figure 1 of the Main text. Are these both performed with purified PDF cells? It looks different from the data in Figure 1, but I am not sure why it does. Fig S2 shows a consistent phase difference across two cycles, but in Figure 1 it is only apparent in the first cycle. Also, 2 biological replicates were superimposed – does that mean each was 48 hours? Or were they 24 hr each and concatenated?

Sorry for the confusion. Fig. S2 is a double plot of the average data from Fig. 1. We've revised this figure with an additional replicate.

Figure S4. Were movement artifacts observed with Hi K? A more traditional graphic feature is to use a bar to indicate the time period when Hi K was perfused (don't just describe it in the figure legend). Was there a difference between expressing one versus two copies of UAS-kir? On a more general note, I was confused by increased kir mitigating the depolarization effects of Hi potassium (line 148-150) - an explanation would be helpful. I thought the membrane potential was essentially controlled by E[Cl] with hi K, so why should additional inward rectifying K channel change that? Also a technical description of the kir transgene is needed - I found none.

We haven't seen big effects on the quantification by movement artifacts. The brains were mobilized by an anchor in the chamber. However, there was still some slight movement in some preps. This is a universal problem and not biased to some particular genotypes. When we quantified, we selected a bigger area to make sure the fluorescent cells don't move out of the selected quantification area.

We changed to a bar to indicate Hi K as suggested.

The Kir experiment has been removed as we are not sure about the mechanism.

Figure S6. Were values between ZT16 and ZT22 significantly different?

It is not significant.

Figure S 9. I was confused by discussion of the sirt2 RNAi stocks – the text (lines 249-252) describes on sick and one ineffective but shows two others; would be cleared up by simply using numerical identifiers for each of the three. Also – why was this data normalized to Rpl32, while other RT-PCR reported (like Figure S6) was normalized 2s rRNA?

This is revised accordingly. We normalized mRNA to rpl32 and miRNAs to 2S rRNA because 2S Rrna is 30 nt long and is co-amplified along with the miRNAs in the assay. It therefore serves better as a control in this case.

Figure S10. Axonal crossing are quantified but there is no indication of statistical analysis.

The statistical analysis is Fig. 6B.

Figure S11. Error bars are shown but N=2.

Correct and thank you. We have eliminated the error bars.

Table S1. Suggest reporting raw values and not the average of two biological replicates.

Revised accordingly.

Reviewer #4 (Remarks to the Author):

In their manuscript "MicroRNA-92a is a Circadian Modulator of Neuronal Excitability in *Drosophila*", Rosbash and colleagues characterize mir-92a as a cycling locus in PDF cells, a key population of circadian pacemakers. They utilize a battery of functional assays, including behavioral activity profiling and neuronal activity measurements, to show that mir-92a suppresses neuronal excitability. Finally, they assign *sirt2* as a functionally relevant target of mir-92a that appears to mediate some of the phenotypes in mir-92a depletion conditions.

I did enjoy reading the paper, and overall I support it for publication once it is appropriately revised. They have used an impressive diversity of sophisticated neurobiological and molecular techniques, which puts this study at the fore of investigations on in vivo miRNA function. The paper is overall easy to follow. However, I have some experimental comments regarding experiments that should be addressed.

1. I find it odd they never use the available KO mutants that were generated by multiple labs and are available at the Bloomington Stock Center. It is possible that null flies do not necessarily have to show the same phenotypes as the tissue specific effects reported, since other behaviors may be masking or even making impossible to carry out the assays, but at least they should try it.

We added experiments with mir-92a null and a *sirt2* amorphic strain. mir-92a shows significant sleep loss (Fig. 3C) and a possible projection abnormality. (We assayed two mir-92a KO strains, generated by two different labs with different genomic backgrounds. One shows shorter but defasciculated projections but the other does not. The results are not shown because they are inconclusive.) Also, no phase shift phenotype was observed with mir-92aKO flies, perhaps because of compensation. The *sirt2* amorphic strain on the other hand shows many defects: fasciculated but overgrowth projections; increase of total sleep and behavioral arrhythmia during DD (Fig S17, S18). These phenotypes are too numerous and complicated to contribute to our story but are included for completion and because of this request.

Also this can help address the contribution of the related mir-92a and mir-92b, since different papers have addressed overlapping in vivo requirements for these miRNAs in *Drosophila* (eg. Yuva-Aydemir Plos Genetics 2015), and a sponge cannot distinguish these isoforms. The above paper indicates that mir-92a and mir-92b are cotranscribed, and thus one may infer they both have rhythmic expression and affect some processes they studied. (For that matter, since these miRNAs are also produced from the *jigr1* gene, one also wonders whether *jigr1* is on the list of cycling loci.

The sponges according to the Van vector lab distinguish between mir-92a and mir-92b. Data from the Griffith lab at Brandeis University show they have similar effects on sleep, but mir-92b is weaker. *Jigr1* is not cycling in PDF cells according to our unpublished data. So it is possible that mir-92a cycling is regulated post-transcriptionally. Similar expression patterns between mir-92a and mir-92b were observed when we performed the 3rd repeat of expression profiling in PDF cells (data not shown).

Or whether it has a function in rhythmic behavior, although I believe this is outside the scope of this study).

According to Yuva-Aydemir Plos Genetics 2015, *jigr1* is auto-regulated. It may therefore be hard to see a phenotype with a knockdown.

In any case, I consider it necessary to confirm at least some of the key behavioral phenotypes and/or anatomical results on PDF cell projections attributed from sponge expression using mutants.

Additional experiments were performed to address this issue (see answer to review #4, question 1)

2. Since the story begins with studying cycling miRNAs in PDF cells, the core circadian pacemakers, it is odd

why there don't seem to be any data regarding rhythmicity when the miRNA is manipulated. This seems like very basic data that they probably generated, but if the results were either positive or negative they should describe the data.

To address this comment, Fig. S9 as well as additional text has been added under the section 'Light pulses at night alter mir-92a levels in PDF cells, and mir-92a levels affect the phase shift response.'

3. The manuscript also rises an interesting consideration: do cycling miRNA levels correspond to cycling miRNA activity? The results and paper are written to suggest so. However, the data are also compatible with developmental defects. Also, previous studies on mir-92a/b have indicated developmental defects in the nervous system. It would be interesting to see if the effects observed for OE and SP are reversible if GAL4 expression ceases. This could give some fruitful information on the correlation of miRNA levels oscillation and miRNA activity oscillation.

For example, the effects from mir-92a and mir-92a-SP are expressed constitutively in PDF cells, and not with temporal control. This could imply that the effects observed in PDF termini are a developmental aberration induced by the manipulation of miRNA levels, rather than an adult specific phenotype, and thus that mir-92a cycling has no consequence on PDF cells. If the hypothesis of the authors is that distinct levels of mir-92a induce different PDF termini conformations, the authors should perform these manipulations strictly in the adult stage, and show its reversibility. I suggest that they add tubGal80ts to the genotype of the OE and SP flies, and show that while raised at 18C, flies show the expected PDF termini at ZT2 and ZT14, but when transferred to 29C, both ZT look alike (fasciculated for OE and defasciculated for SP). Then, upon transferring the flies back to 18C, the PDF termini cycle normally again.

Since temperature affects projection morphology, we address the adulthood specificity with PDF-GSG (Fig. S4, S13).

4. Sirt2. There are published sirt2 mutant flies and even sirt2 gain of function flies. Similar to the mir-92a studies, the role of sirt2 is not tested with genuine mutants, which are available at the Bloomington Stock Center. I would like to see this confirmed.

While we are unable to locate the gain of function strain, we performed experiments with the *sirt2* amorphic strain and observed some similar phenotypes to the *sirt2* RNAi approach (Fig. S16, S17).

Minor comments:

miRNA quantification:

.- Fig. 1A: miRNA profile in the two light periods, and in the two dark phases shown seem very different in terms of levels, peaks, and tendency. Also, in fig 1C, in the absence of per protein, miRNA levels amplitude of variation is as high as observed under wt conditions in Fig.1A. Why are the authors convinced then, that fig.1A is unambiguously showing cycling of miRNA levels, and not just showing some unspecific variation? Also, the mir-92a levels in Fig.1A and in Fig. S2 look quite different, with the latter showing almost an idealized regular up and down pattern, much more regular than the former. Showing the expression range from the replicate data would be useful to interpret.

We did an additional biological replicate to address this point. Fig. S2 was the average of two replicates double plotted, and it is now the average of three replicates. Since the samples are from sorted neurons, which are manually sorted, the variations are inevitably larger than preps from whole heads or other tissues that are easier to prep.

Neuronal excitability:

.- The genotype of the control is not stated for the OE expression in fig 2B, while it is stated in fig 1A. The controls here should be the Gal4+arc light and UAS+ArcLight in both experiments.

Sorry but we don't understand the comment. There is no control used in Fig. 1A.

.- What is the ZT of the brains used in the ArcLight experiment? Since the time of day is determinant for neuronal excitability, I assume that all the brains were synchronized, although this is not specified in the manuscript. This should be clarified. If I understand, for the OE data, both control and experimental should be ZT2 (max excitability in the controls and reduced in the OE). For the SP experiment, it should be done in ZT14 (min excitability for the control, increased in the experimental). Is this the design used? If another design was used, then the authors should clarify.

The experiments were performed at ~ ZT6-8. This is because the experiments required light to set up, dissect, anchor the brains to the chamber, laser from the microscope etc. To minimize the light effect on arclight signal, we therefore chose a time point during the light period. But it is entirely possible that a bigger effect with mir-92aSP will be observed if the experiments were performed at night.

CaLexA:

.- Line 164: "especially during the morning". I am not sure if I interpreted right, but it seems the main reduction is more apparent in the second half of the light phase, that would correspond to the afternoon or evening, but in any case the morning (Fig2. C OE panel).

Sorry for the confusion, morning indicated the light period (Zt0-12) here. We revised it according.

.- mir-92a SP levels show an acute peak at night. Nevertheless, if neuronal excitability depends on miR-92a levels (and correlate to PDF conformations), and these are kept uniform in the KD genotype, shouldn't the excitability be constant as well? Why does it clearly decays in the beginning of the night and during the morning instead of showing a plateau? Same for OE experiments.

Mir-92a may not have such a big effect on neuronal excitability. That's why we said it modulates neuronal excitability. Neuronal excitability should be still cycling in the flies, since the rhythmicity is also not affected.

Sirt2:

.- Line 230: They say "mutant flies" and that is not correct, they mean KD

Revised accordingly.

.- The conclusion raised in the last sentence is overstated. From the in vitro sensor experiment, they can only state that mir-92a suppresses sirt2 expression in vitro by binding to the mentioned site. They cannot be sure if it does so in vivo, for that they need to delete that binding sites in live animals and check sirt2 expression in those conditions.

Revised accordingly.

REVIEWERS' COMMENTS:

Reviewer #1 (Remarks to the Author):

The authors have responded well to my comments and I think that this study should be published.

Reviewer #2 (Remarks to the Author):

The authors have satisfactorily responded to my concerns.

Reviewer #3 (Remarks to the Author):

The authors have an excellent job in responding to the many concerns and criticisms from the previous round of review. This manuscript clearly describes complicated interactions between the clock, light and the mir-92a pathway. It uses a number of different assays well (but see specific concerns below) to make an overall impressive case for mir-92a controlling sirt2 to affect circadian pacemaker and DA neuron physiology.

Remaining Concerns

1. the principal remaining concern is the lumping of descriptions concerning PDF neurons between large and small LNV. The small and large are quite different in their physiology, including some rhythmic properties. Is it certain or clear that the mir-92a normally works in both cell types? According to my reading, the GCaMP6 measures were from large LNV (or both?), The in vivo Calnexa likely from the large LNV, the axon fasciculation measures were from small LNV, and I am not sure about the Arlight measures. I recommend the authors make clear which cell type is being studied in the various sections and also make sure to take this difference into account in their Discussion of the role(s) of mir-92a, and p 7 not simply speak of the role in "PDF neurons".

2. In Figure 1 and Suppl Figure 2, the 24 hr PCR results are double plotted: whereas Suppl Figure 2 shows an authentic 48 hr series of data points. Double plotting is used as a convention for inspecting many days/weeks of rhythmic activity and to more easily perceive trend-lines of change. I am not supportive of using it in this Figure because I see no justification for doing so, beyond creating a stronger appearance of daily rhythmicity. The single cycle of 24 hr data appears rhythmic as described.

3. I recommend deleting S Figure 5 Average GCaMP6 fluorescence levels of PDF cell bodies (I-LNVs) because it does not advance the argument made concerning the role of mir-92a as described. The authors write on p 9:

"Changes are statistically insignificant by two-way ANOVA.

GCaMP6 live imaging of PDF neurons may show an altered responsiveness to nicotine in mir-92aOE and mir-92aSP flies.

Changes among genotypes are not statistically significant ...Nonetheless, the results are consistent with those from the KCl experiments, suggesting that adult mir-92a expression suppresses PDF cell responsiveness to environmental stimuli."

I disagree – trends and non-significant results do not support a hypothesis. This experiment should be reported as a negative result, or removed.

Smaller points

1. p 7, line 97

(data not shown) – is this still permitted?

p 7, l 104

“manuscript” - once a paper is published, is it still a manuscript?

2. Figure 2B – Arclight: Left panels show the average responses of 8 PDF neurons of the indicated genotypes-

8 neurons in how many total brains?

Small or large LNv? See point #1 above)

Were the ROIs over cell bodies or processes?

“Interestingly, mir-92aOE significantly decreased the response, and mir-92aSP consistently but insignificantly increased the response (Fig. 2B).”

I think there is no need to feature insignificant results: the only consistent results worth interpreting are the significant ones.

3. Arclight measurements (Figure 2) were performed ZT6-8. However, GCaMP6 measurements (S Figure 5) “were performed between ZT18 – 22 (when endogenous mir-92a levels in PDF cells are high), and w1118 and mir-92aOE between ZT6 – 10 (when endogenous mir-92a levels in PDF cells are low).”

Why the difference?

Reviewer #4 (Remarks to the Author):

I am satisfied with the author's revision and their added experiments.

Reviewer #3 (Remarks to the Author):

The authors have an excellent job in responding to the many concerns and criticisms from the previous round of review. This manuscript clearly describes complicated interactions between the clock, light and the mir-92a pathway. It uses a number of different assays well (but see specific concerns below) to make an overall impressive case for mir-92a controlling sirt2 to affect circadian pacemaker and DA neuron physiology.

Remaining Concerns

1. the principal remaining concern is the lumping of descriptions concerning PDF neurons between large and small LNV. The small and large are quite different in their physiology, including some rhythmic properties. Is it certain or clear that the mir-92a normally works in both cell types? According to my reading, the GCaMP6 measures were from large LNV (or both?), The in vivo Calnexa likely from the large LNV, the axon fasciculation measures were from small LNV, and I am not sure about the Arlight measures. I recommend the authors make clear which cell type is being studied in the various sections and also make sure to take this difference into account in their Discussion of the role(s) of mir-92a, and p 7 not simply speak of the role in "PDF neurons".

Thanks for this suggestion and requesting this clarification. We agree completely that we should have specified which neurons were measured in each experiment, and we have now done this in the "final" manuscript. Please note: it is very likely that both large and small LNVs are affected by mir-92a, and we have added this point to the Discussion.

2. In Figure 1 and Suppl Figure 2, the 24 hr PCR results are double plotted: whereas Suppl Figure 2 shows an authentic 48 hr series of data points. Double plotting is used as a convention for inspecting many days/weeks of rhythmic activity and to more easily perceive trend-lines of change. I am not supportive of using it in this Figure because I see no justification for doing so, beyond creating a stronger appearance of daily rhythmicity. The single cycle of 24 hr data appears rhythmic as described. [Or you can change it Xiao; you decide.]

The reviewer is correct that double plotting is commonly (and was originally) done for long-term behavioral monitoring. However, this strategy is not unprecedented for cycling RNA profiles. Instead of plotting three repeats as consecutive multiple days, data are sometimes averaged and then double plotted. This strategy is especially useful when only a trough is visible, i.e., values are elevated at the beginning and the end of the day. Double plotting then "creates" a visible peak. The reviewer agrees that the single cycle appears rhythmic. So there is no harm done by double plotting, i.e., it is then almost an aesthetic choice, isn't it?

3. I recommend deleting S Figure 5 Average GCaMP6 fluorescence levels of PDF cell bodies (l-LNVs) because it does not advance the argument made concerning the role of mir-92a as described. The authors write on p 9:

"Changes are statistically insignificant by two-way ANOVA.

GCaMP6 live imaging of PDF neurons may show an altered responsiveness to nicotine in mir-92aOE and mir-92aSP flies.

Changes among genotypes are not statistically significant ...Nonetheless, the results are consistent with those from the KCl experiments, suggesting that adult mir-92a expression suppresses PDF cell

responsiveness to environmental stimuli.”

I disagree – trends and non-significant results do not support a hypothesis. This experiment should be reported as a negative result, or removed.

We respectfully disagree with this recommendation. If there were little or no additional data supporting the hypothesis, then we would agree. However, with all of the multiple lines of positive evidence showing that mir-92a regulates excitability, the trends are not without interest. Why else keep the experiment here? Someone may decide to repeat it with a larger N; the trend may then become significant, or the trend may disappear; either would be important to know and therefore encouraging further study is in our interest. In addition, having the experiment here rather than deleting it facilitates using this strategy again in the context of *sirt2* later in the paper. In that case, the effects are significant and therefore entirely positive. Note that we were honest about this result and now propose to be even more explicit and describe it as a “negative result” as suggested by the reviewer.

Smaller points

1. p 7, line 97

(data not shown) – is this still permitted?

p 7, l 104

“manuscript” - once a paper is published, is it still a manuscript?

Thanks for the questions. The statement ‘data not shown’ has been deleted and ‘manuscript’ is permitted to use.

2. Figure 2B – Arlight: Left panels show the average responses of 8 PDF neurons of the indicated genotypes-

8 neurons in how many total brains?

Small or large LNv? See point #1 above)

Were the ROIs over cell bodies or processes?

“Interestingly, mir-92aOE significantly decreased the response, and mir-92aSP consistently but insignificantly increased the response (Fig. 2B).”

I think there is no need to feature insignificant results: the only consistent results worth interpreting are the significant ones.

We have added description regarding the first couple questions.

Regarding the last point, we respectfully disagree. We have done everything both ways, i.e., overexpression and sponge. It would be disingenuous to report only one. And since we show both directions for all positive assays, it makes sense to still show both in the couple of assays that do not give consistent positive results in the two directions. So we are just reporting what we observe, namely, a consistent but insignificant trend. Readers can then decide for themselves how they wish interpret the data. Moreover and as mentioned above for the one other experiment with insignificant trends in the right direction, someone may decide to repeat the experiment with a larger N and confirm or reverse the trend.

3. Arclight measurements (Figure 2) were performed ZT6-8. However, GCaMP6 measurements (S Figure 5) "were performed between ZT18 – 22 (when endogenous mir-92a levels in PDF cells are high), and w1118 and mir-92aOE between ZT6 – 10 (when endogenous mir-92a levels in PDF cells are low). "

Why the difference?

We originally used ZT6 - 8 to avoid effects from light. Since the results from mir-92aSP flies with Arclight were negative at this ZT time, we decided to use ZT18 - 22 for the GCaMP6 experiment to increase the chances that a significant effect could be observed.